**Resource**

# Detailed mapping of the complex fiber structure and white matter pathways of the chimpanzee brain

Cornelius Eichner [1] ✉, Michael Paquette[1], Christa Müller-Axt [1,2], Christian Bock [3], Eike Budinger[4,5], Tobias Gräßle[6], Carsten Jäger[7,8], Evgeniya Kirilina [7,9], Ilona Lipp[7], EBC Consortium*, Markus Morawski[8], Henriette Rusch[8], Patricia Wenk[4], Nikolaus Weiskopf [7,10], Roman M. Wittig [11,12,13], Catherine Crockford[11,12,13], Angela D. Friederici [1] & Alfred Anwander [1]

Long-standing questions about human brain evolution may only be resolved through comparisons with close living evolutionary relatives, such as chimpanzees. This applies in particular to structural white matter (WM) connectivity, which continuously expanded throughout evolution. However, due to legal restrictions on chimpanzee research, neuroscience research currently relies largely on data with limited detail or on comparisons with evolutionarily distant monkeys. Here, we present a detailed magnetic resonance imaging resource to study structural WM connectivity in the chimpanzee. This open-access resource contains (1) WM reconstructions of a postmortem chimpanzee brain, using the highest-quality diffusion magnetic resonance imaging data yet acquired from great apes; (2) an optimized and validated method for high-quality fiber orientation reconstructions; and (3) major fiber tract segmentations for cross-species morphological comparisons. This dataset enabled us to identify phylogenetically relevant details of the chimpanzee connectome, and we anticipate that it will substantially contribute to understanding human brain evolution.

The ability to drive scientific progress, attain virtuosity in fine arts or to use language are examples of human-specific skills. The basis for these remarkable capacities lies in the human brain's complex structural and functional architecture.

To date, however, it is not well understood how the human brain structure developed throughout evolution. Direct comparisons between brains of modern humans and their extinct evolutionary ancestors are inherently impossible. Consequently, human evolutionary neuroscience profits from comparisons between humans and nonhuman primates.

Primate brain evolution is characterized by an increase in brain size and a marked increase in the proportion of white to gray matter[1–3].

WM fiber connections enable interactions between neurons of different cortical and subcortical gray matter areas and are the central neurobiological basis for the mastery of complex cognitive abilities among primates. Technical and methodological advances in magnetic resonance imaging (MRI) allow mapping these fiber tracts throughout the brain noninvasively.

Diffusion MRI (dMRI) yields information on microstructural WM tissue characteristics and structural connections in humans and nonhuman primates[4]. A substantial challenge in dMRI tractography is the accurate reconstruction of complex WM architecture, which entails crossing fibers in nearly every region of the brain[5]; however, accurately resolving WM microstructure[6] and connectivity is of keen interest to

**Fig. 1 | High-resolution MRI data quality. a**, Whole-brain color-coded FA reconstruction of the acquired high-resolution (500 µm isotropic) postmortem chimpanzee dMRI dataset. The color indicates the tissue orientation and the brightness encodes the anisotropy. S, superior; A, anterior; P, posterior; R, right; I, inferior; L, left. **b**, Zoomed regions of DTI (top row) and FLASH MR microscopy at 150 µm isotropic resolution (bottom row). Anatomical labels are provided on the FLASH MR microscopy data. ll, lateral lemniscus; ml, medial lemniscus; cwm, cerebellar white matter; cau, caudate; gpe, external globus pallidus; th, thalamus; gpi, internal globus pallidus; ac, anterior commissure; pt, putamen; ac, anterior commissure; or, optic radiation; lgn, lateral geniculate nucleus. Abbreviations are defined in Supplementary Table 1. **c**, Exemplary high-resolution human in vivo dMRI DTI data with 1-mm isotropic resolution from the 7T Human Connectome Project (subject 126,426).

better understand higher cognitive functions, such as language[7] and neurological diseases[8].

Comparisons of structural WM brain connections in humans with homologous pathways in apes and monkeys offer an opportunity to investigate the neurobiology of brain evolution[9–13]. In these efforts, postmortem dMRI can play a unique role by providing data of high quality[14–16]. Several macro- and microstructural characteristics indicate that brains of great apes are closer to human brains than to those of other primate clades, such as Cercopithecidae (for example, macaque monkeys)[17,18]. Thus, a thorough anatomical description of brains of great apes is key to understanding the evolution of the human brain.

However, ethical concerns related to primate research severely limit the options for neuroimaging research involving great apes. Most countries have adopted moratoria on conducting invasive chimpanzee research, including MRI performed under anesthesia[19]. Consequently, current evolutionary neuroscientific knowledge is restricted mainly to comparisons of humans with evolutionary distant monkeys, such as macaques[20]. As for comparisons with great apes, research is limited to data collected before the moratorium[11,12], for example The National Chimpanzee Brain Resource (https://www.chimpanzeebrain.org). These in-vivo neuroimaging data from chimpanzees, acquired many years ago, are limited in image resolution and neuroanatomical

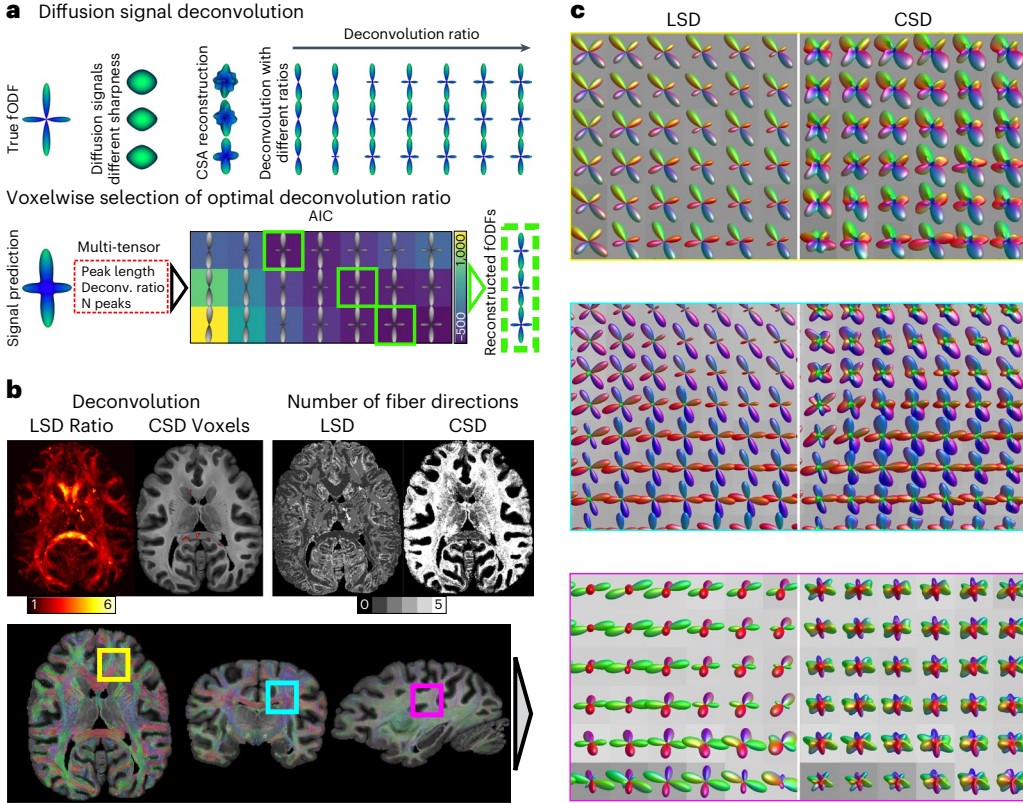

**Fig. 2 | Local spherical deconvolution. a**, Illustration of LSD reconstruction. Ground-truth fODF might be embedded in different microstructural environments, yielding different dMRI signals. The constant solid angle (CSA) approach reconstructs diffusion orientation distribution functions (dODF) for each voxel. Each voxel is deconvoluted using kernels of various anisotropy. The AIC determines the best ratio using local multi-compartment forward modeling. **b**, Comparison between LSD and CSD reconstruction of optimal deconvolution between LSD and CSD (left). LSD utilized various kernel sharpness across the brain. CSD focused on cc voxels for deconvolution. Comparison of identified NuFO between LSD and CSD (right). **c**, Zoomed comparison between LSD and CSD fODFs in different brain regions with crossing fiber anatomies: yellow indicates axial frontal crossing between ifof and cc, cyan indicates coronal crossing between cc and cst and magenta indicates sagittal crossing between af and cc.

accuracy. Previous attempts to acquire postmortem chimpanzee dMRI data suffered from limited resolution and low signal, impeding accurate reconstructions of complex WM fiber architecture and connectivity[13,21].

This study aimed to provide a detailed morphological description and characterization of the brain WM pathways in the chimpanzee, one of our closest living evolutionary relatives. To this end, we obtained whole-brain dMRI data from a single adult chimpanzee, which was euthanized for medical reasons. The sample was obtained for the 'Evolution of Brain Connectivity Project', an international research effort exploring the evolutionary trajectory of the human brain[22,23]. Through multiple days of optimized scanning at ultra-high field strength, we acquired dMRI data with unprecedented image resolution. To take full advantage of the data quality attained, we developed an optimized dMRI reconstruction technique capable of reliably resolving complex fiber configurations. Here, we present the first high-resolution whole-brain atlas of WM fiber pathways in the chimpanzee. We release this MRI dataset to facilitate efforts to understand the evolution of the human brain.

## Results

### Ultra-high-resolution MRI data

We acquired a whole-brain chimpanzee dMRI dataset at 500 μm isotropic voxel size on a 9.4T preclinical MRI system (Fig. 1a,b). Microscopical and histological assessment revealed a well-preserved brain cyto- and myeloarchitecture, showing clear myelin layers[24] (Supplementary Note 1). We acquired data over multiple days, using an optimized segmented three-dimensional (3D) EPI sequence, yielding a high mean

signal-to-noise ratio of 83.6 (Supplementary Note 2), low distortions and low blurring. We leveraged an optimized diffusion-weighting of $b = 5,000$ s mm$^{-2}$ (Supplementary Note 3). A test–retest assessment revealed high stability of the dMRI acquisition and reconstruction (Supplementary Note 2). We show color-coded fractional anisotropy (FA) images, derived from the diffusion measurements (Fig. 1a,b). A breakdown of the main factors contributing to the high data quality is provided in Supplementary Note 3.

We acquired fast low-angle shot (FLASH) data with identical alignment and voxel size as the dMRI data, but with varying contrasts (PD-weighted to $T_1$-weighted). Moreover, we obtained a complementary magnetic resonance (MR) microscopy dataset with an isotropic voxel size of 150 μm (Fig. 1b, bottom).

The present dMRI and MR microscopy data allowed us to resolve anatomical details that had previously been described in histological data but remained hidden in earlier MRI datasets (Fig. 1b): In the brainstem, fine anatomical details of the pons, such as the pontine nuclei (pn) or the pontine crossing tract (pct), are well visible. In addition, the demarcation of the corticospinal tract within the pons and the lateral and medial lemnisci can be seen (Fig. 1b). In the cerebellum, the high-resolution data reveal the finely branched cerebellar foliate (cf) structure and the cerebellar WM (Fig. 1b). In the striatum, anatomical details of Edinger's comb between the putamen and caudate are well visible. As in other primates, WM fibers of the internal capsule (ic) intersect with striatal cell bridges (sb) in this region (Fig. 1b). In the hippocampus, the data resolution allows studying the rolled structure of the dentate gyrus (dg) (Fig. 1b). This level of anatomical detail is far

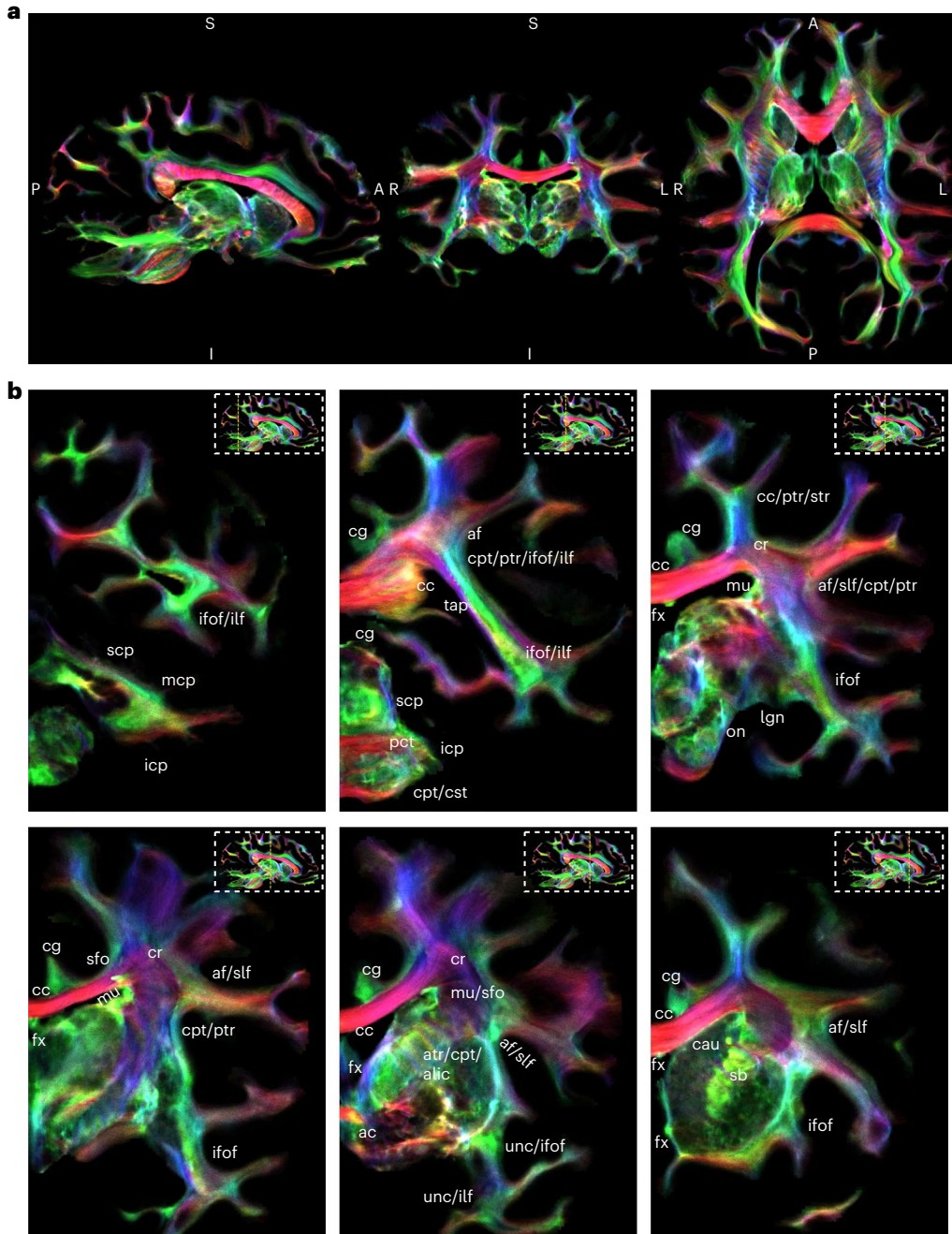

**Fig. 3 | Whole-brain tractography. a**, Whole-brain track-density reconstruction of WM pathways in the chimpanzee brain. Overview of the whole chimpanzee brain on a color-coded track-density image (TDI). **b**, Labeled series of sequential coronal slices of the track-density data with anatomical labels, zoomed to the left hemisphere. Abbreviations are defined in Supplementary Table 1. The orientation of the different fiber tracts is color-coded. In AP orientation (green), the ifof, the ilf, the cingulum (cg) and the fornix (fx) can be delineated. In LR orientation (red), the main inter-hemispheric pathways such as the cc and the ac can be seen. In inferior–superior (IS) orientation (blue), the cst, tapetum (tap), the anterior, posterior and superior thalamic radiation (atr, ptr and str), as well as the corticopontine tract (cpt) are visible. In the cerebellum, fascicles such as the inferior cerebral peduncle (icp), mcp and superior cerebral peduncle (scp) are detectable.

beyond the current state of the art for human in vivo dMRI data (for example, 7T dMRI[25]; Fig. 1c) as well as previously acquired postmortem chimpanzee data (Supplementary Note 4).

## Local spherical deconvolution

We reconstructed the local WM fiber structure using the acquired high-resolution dMRI data. Simple dMRI models such as diffusion tensor imaging (DTI) do not support detecting multiple fiber orientations within a voxel; however, accurate and stable modeling of crossing fibers is paramount for an anatomically meaningful reconstruction of structural brain connectivity.

The most widely used method to estimate crossing fibers is constrained spherical deconvolution (CSD)[26]. CSD relies on one single-fiber response function, called the deconvolution kernel, to compute the fiber orientation distribution functions (fODF) in the entire WM. This kernel is generally estimated from brain regions with strongest anisotropy; however, the appropriate kernel choice is crucial, as it directly impacts the estimated number of fiber orientations (NuFO) and may

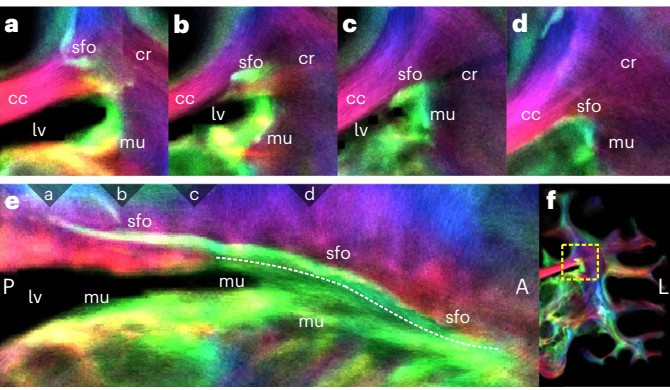

**Fig. 4 | Superior fronto-occipital fascicle (sfo) and the subcallosal fascicle of Muratoff (mu). a–d**, Coronal view of sfo and mu, splitting paths in the corpus callosum. **e**, Sagittal view of jointly running mu and sfo in AP orientation. The triangles at the upper margin indicate the positions of the coronal slices (**a–d**). After segregating (**c**), mu runs along the ventricular wall into the striatum. In contrast, the sfo runs dorsally toward the cortex. **f**, Left hemisphere of the brain for spatial orientation. Abbreviations are defined in Supplementary Table 1.

indicate erroneous directions[27]. Initial results on tract-specific microstructural properties challenge CSD assumptions and call for adapted kernels in different brain regions[27,28].

To overcome nonoptimal kernel selection, we developed an optimized fiber orientation reconstruction algorithm, termed local spherical deconvolution (LSD), which uses a voxel-specific optimal deconvolution kernel adapted to local tissue properties. It accounts for different microstructural environments of axons and best represents the acquired dMRI data. The optimal kernel selection leverages the Akaike information criterion (AIC) (Fig. 2a). A test–retest assessment showcased the stability of LSD reconstruction parameters and estimated fiber orientations (Supplementary Note 2).

For CSD, the established kernel selection method chose highly anisotropic corpus callosum (cc) voxels (Fig. 2b). As a result, CSD relied on a kernel with anisotropy too high to adequately represent the dMRI signal in most regions of the brain, resulting in inflated NuFO values (Fig. 2b) of up to five orientations per voxel.

In contrast, LSD identified optimal deconvolution kernels throughout the brain (Fig. 2b). The kernel shape was optimized for every voxel and indicated spatially varying diffusion anisotropy throughout the brain (Fig. 2b). This approach led LSD to estimate more anatomically plausible NuFO values (Fig. 2b).

In comparison to CSD, LSD fODFs seemed sharper and more ordered, resembling known human anatomy (Fig. 2c). This is particularly evident in crossing fiber regions, such as (1) the intersection between anterior thalamic radiation and genu of the cc; (2) the crossing between the cc body and the corticospinal tract (cst); and (3) the crossing region between the superior-longitudinal fascicle (slf)/arcuate fascicle (af) and the transcallosal tracts (Fig. 2c). In the af region, LSD confirmed the anatomically known association/projection fibers and the projections of the cc present in the primate lineage[29,30]. In contrast, CSD seemed to suffer from the overly sharp reconstruction kernel, resulting in a fragmentation of reconstructed fiber orientations in the anterior–posterior (AP) direction (Fig. 2c).

We thoroughly validated the LSD reconstruction accuracy against the ground truth from numerical simulations, known human anatomy and myelin histology (Supplementary Note 5). This validation also entails a detailed comparison and benchmark of LSD against CSD, as well as an analysis of LSD reconstruction stability against the acquired number of diffusion directions.

## Visualization of WM structures

**General description.** We studied the chimpanzee's WM brain organization with its major structures in the left hemisphere (Fig. 3). The large fiber tracts, also known from humans[31,32], are immediately apparent.

For example, in AP orientation (green), the slf and the af are visible. Both slf and af run largely parallel, separating at the temporoparietal junction. The slf makes a lateral turn and connects with the supramarginal gyrus (smg). The af takes a descending turn to connect to the planum temporale and primary auditory cortex within the superior temporal gyrus. This tract has been repeatedly linked to the evolution of human language[9,11,33]. The slf and af seem strongly interspersed with transcallosal connections in the left–right (LR) orientation. Compared to CSD, the LSD algorithm accurately reconstructed both fiber orientations within these critical regions (Fig. 2b).

**Mapping the superior fronto-occipital fascicle.** In addition to mapping the chimpanzee's principal fascicles, the high-resolution data allow insights into subtle details of primate WM structure, such as the superior fronto-occipital fascicle (sfo) connecting the parietal and occipital lobe with frontal regions and the subcallosal fascicle of Muratoff (mu) (Fig. 4) connecting the striatum with regions in the frontal lobe. Both pathways were previously identified in monkeys using invasive tract-tracing methods[34]. In humans, however, the morphology or existence of these pathways remains controversial[35,36].

In this dataset, both the sfo and mu can be clearly identified and distinguished. In the frontal lobe, they run closely parallel and separate at the level of the lateral ventricle (lv) (Fig. 4). The sfo then continues dorsally toward the cortex. Conversely, the mu continues along the ventricular wall into the striatum. This suggests that a prominent sfo also exists in chimpanzees. Considering that the sfo is substantially reduced or even absent in humans, the regression of this tract may be an evolutionary brain change specific to humans.

**Details of inter-hemispheric connections in the chimpanzee.** The present data allow a detailed investigation of inter-hemispheric connections in the chimpanzee brain. At high image resolutions, these connections reveal fascinating details, such as the laminar structure of the cc (Fig. 5a). Early histological brain preparations have shown that fiber arrangements in the cc extend along radially running laminae[37]. This has recently also been demonstrated in living humans using 7T-FLASH data[38]. In contrast to previous dMRI acquisitions, the present data allow the direct visualization of the cc lamellae in the chimpanzee brain. (Fig. 5a).

The present dataset provides sufficient spatial resolution to map all primary inter-hemispheric connections in the chimpanzee (Fig. 5b). At first glance, principal inter-hemispheric commissures, such as the cc, ac and pc stand out; however, several additional, typically indiscernible structures can be observed, such as the middle commissure (mc) and the habenular commissure (hc). We did not resolve the ventral hippocampal commissure (vhc) in the present chimpanzee dataset. In contrast, the vhc was recently documented in a postmortem study in the marmoset monkey[14]. As this study used a comparable relative voxel size to our approach, the observed discrepancy may indicate an evolutionary regression of the vhc between apes and monkeys.

## Reconstruction of chimpanzee brain fiber pathways

The high quality of the acquired dMRI data and the LSD reconstruction allows us to render principal fascicles with an excellent level of anatomical detail (Fig. 6): We discern the dorsal longitudinal fiber pathways, including the slf I–III and the af, with only the af bending posteriorly into the posterior superior temporal gyrus (stg). Frontally, the af connects to the inferior frontal gyrus (ifg), like the slf III (Fig. 6a). We show the inferior fronto-occipital (ifof) and the uncinate (unc) fascicle, connecting the frontal cortex with the occipital cortex and the temporal pole, respectively (Fig. 6b). We show the two main inferior frontal pathways

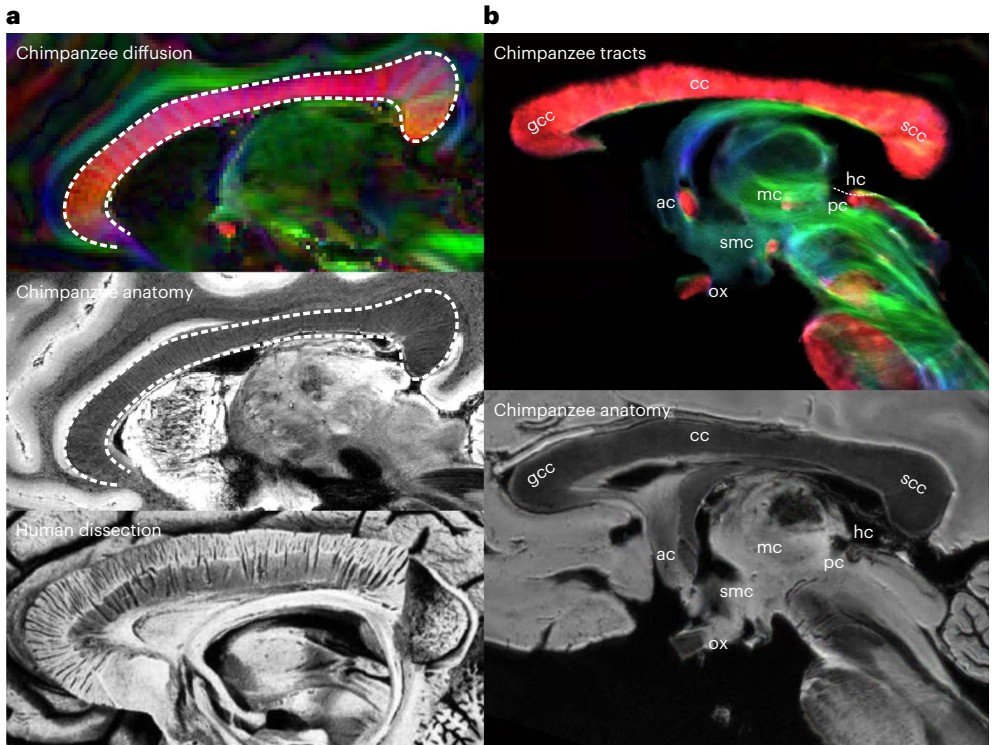

**Fig. 5 | Inter-hemispheric connections. a**, The laminar structure of the chimpanzee cc is visible in the DTI data (top, sagittal plane), in which the cc shows lamina with a tissue orientation that is orthogonal to the LR orientation (red) of the cc axons. The outline of the cc is indicated by a white dashed line. This sheet-like arrangement is otherwise mainly known from human dissection data (bottom, reproduced from Ludwig and Klingler, 1956, permission granted by publisher). **b**, The numerous inter-hemispheric pathways of the chimpanzee brain can be observed in sagittal view. Anatomical labels are provided. Definitions of abbreviations are provided in Supplementary Table 1. gcc, genu of the corpus callosum; scc, splenium of the corpus callosum; smc, supramammilliary commissure.

of the temporal lobe: the inferior longitudinal fascicle (ilf) and the medial longitudinal fascicle (mdlf), which connect the temporal with the occipital lobe (Fig. 6c), the cingulum, which connects the medial frontal lobe and cingulate gyrus with the medial parietal lobe and the entorhinal cortex (Fig. 6d), the cst and middle cerebellar peduncle (mcp) (Fig. 6e). The crossing between the mcp and cst in the pons is well resolved, as also visible in the directionally encoded axial slice (Fig. 1c). For individual tract reconstructions and anatomical labels refer to Supplementary Note 6.

## Discussion

Understanding the evolutionary origin of the human brain and its connectivity is a pressing question in neuroscience. To this end, functional and structural comparisons are made between the brains of humans and other species, particularly nonhuman primates, to decipher the evolutionary path of the brain through comparisons with evolutionary relatives. Critical for unraveling the origins of human abilities is the comparison with the chimpanzee, one of our closest living relatives, with which we shared a last common ancestor about 7 million years ago[39].

Previously acquired chimpanzee neuroimaging data are scarce and suffer from limited image resolution and anatomical accuracy. To bridge this gap, we here present a high-quality postmortem dMRI and MR microscopy resource. Following previous releases of high-quality postmortem diffusion resources from marmoset monkeys[14] and macaques[16], our data provide a detailed insight into the brain connectivity of our closest living evolutionary relatives. As such, our data complement current evolutionary brain research by providing the opportunity to formulate detailed hypotheses and validate findings from large data sources such as the National Chimpanzee Resource. The raw and processed data, including the tractography results, are openly shared with the research community (https://openscience.cbs.mpdl.mpg.de/ebc).

The primate brain connectome is characterized by fiber tracts of various sizes. For a comprehensive understanding of WM, a detailed representation of all fascicles, including fine tracts, is crucial; however, noninvasive imaging methods such as conventional in vivo dMRI suffer from low image resolution and fail to reveal the subtle details of brain connectivity. Other methods to identify fiber connections, such as invasive in vivo tract-tracing or postmortem polarized-light-microscopy (PLI)[40], are either not feasible in chimpanzees (tracing) or face considerable challenges in the reconstruction of spatial 3D images/representations (PLI). Accordingly, postmortem dMRI proves a valuable alternative for identifying brain pathways that can achieve unprecedented image resolutions compared to in vivo MRI[14,15,41].

The dMRI resource presented here offers several advantages over previous studies of chimpanzee brain connectivity[9,12,13]. Besides the substantially higher resolution, these advantages include improved microstructural diffusion contrast, realistic representations of fiber crossings and a higher specificity of the reconstructed fiber tracts. Even with large fascicles, these data enable more accurate and detailed representations of the fiber projections on the cortical surface[14].

For a high-quality reconstruction of chimpanzee WM connections, we have developed an optimized fiber orientation reconstruction algorithm (LSD). In contrast to previous reconstruction approaches such as CSD, LSD computes matching deconvolution kernels in each voxel separately to reconstruct WM fiber orientations with high fidelity. LSD increased the neuroanatomical precision of the reconstructed chimpanzee brain connectome, while also accounting for complex WM architectures such as crossing fibers.

The combination of high-resolution imaging and optimal fiber reconstruction provides insights into the neuronal connectivity

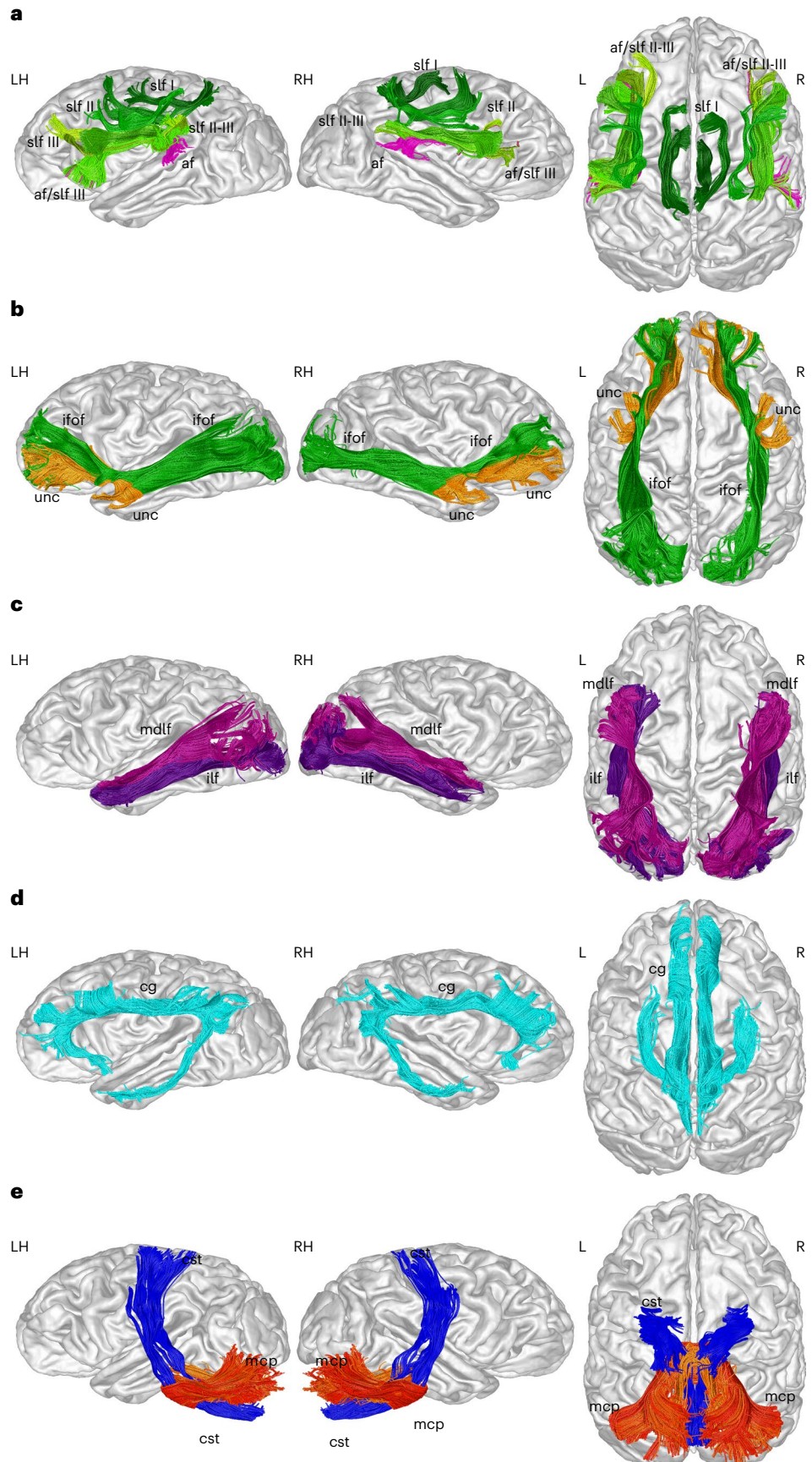

**Fig. 6 | Fiber tract segmentations. a**, Tractography reconstructions of the slf I–III (green) and the af (magenta). Note that the frontal branch of the af is concealed by the slf III. **b**, Tractography reconstructions of the ifof (green) and the unc (orange). **c**, Tractography reconstructions of the mdlf (fuchsia) and the ilf (purple). **d,e**, Tractography reconstruction of the cg (cyan) (**d**), cst (blue) (**e**) and the mcp (red). Abbreviations are defined in Supplementary Table 1.

of the chimpanzee brain. For instance, we were able to observe a clear separation between the frontal brain fascicles sfo and mu. The existence of the sfo is controversial in humans[35]. Our results indicate the existence of the sfo in chimpanzees, thereby placing this tract evolutionarily closer to humans than previously thought. If the sfo has regressed in humans since the last common ancestor with chimpanzees, this would be a unique evolutionary development in humans.

Complex WM architecture, such as crossing fibers, may not only complicate reconstructions of smaller tracts but also their exact cortical projections. Here, the af is of particular interest, as it constitutes a central component of the human language network. Its evolutionary change has been considered essential in explaining the emergence of human language ability[9,11]. Our high-resolution data show that the chimpanzee af is crossed by strong transcallosal tracts. It did not show a connection of comparable strength reaching the middle and inferior temporal gyri as compared to humans. This result supports the hypothesis that the human af, targeting the middle temporal gyrus, may be critical for language processing.

In summary, the present ultra-high-resolution dataset allows a detailed description of the brain organization of the closest living evolutionary relatives of humans, the great apes. Only through detailed inter-species comparisons, can we trace the trajectories of brain evolution. Several considerations make this endeavor a matter of utmost urgency. In vivo imaging experiments on great apes are banned worldwide due to ethical and legal concerns. Simultaneously, all great ape species are threatened by extinction due to poaching and the destruction of their natural habitat[42]. This gradually closes the door to understanding the evolution of the human brain and the emergence of human-specific behavioral traits.

## Online content

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

[1]Department of Neuropsychology, Max Planck Institute for Human Cognitive and Brain Sciences, Leipzig, Germany. [2]Faculty of Psychology, TU Dresden, Dresden, Germany. [3]Alfred Wegener Institute Helmholtz Centre for Polar and Marine Research, Bremerhaven, Germany. [4]Leibniz Institute for Neurobiology, Combinatorial NeuroImaging Core Facility, Magdeburg, Germany. [5]Center for Behavioural Neurosciences, Magdeburg, Germany. [6]Ecology and Emergence of Zoonotic Diseases, Helmholtz Institute for One Health, Helmholtz Centre for Infection Research, Greifswald, Germany. [7]Department of Neurophysics, Max Planck Institute for Human Cognitive and Brain Sciences, Leipzig, Germany. [8]Paul Flechsig Institute - Centre of Neuropathology and Brain Research, Medical Faculty, University of Leipzig, Leipzig, Germany. [9]Center for Cognitive Neuroscience Berlin, Free University Berlin, Berlin, Germany. [10]Felix Bloch Institute for Solid State Physics, Faculty of Physics and Earth Sciences, Leipzig University, Leipzig, Germany. [11]Department of Human Behavior, Ecology and Culture, Max Planck Institute for Evolutionary Anthropology, Leipzig, Germany. [12]Tai Chimpanzee Project, Centre Suisse de Recherches Scientifiques, Abidjan, Côte d'Ivoire. [13]The Ape Social Mind Lab, Institut des Sciences Cognitives Marc Jeannerod, Lyon, France. *A list of authors and their affiliations appears at the end of the paper. ✉e-mail: ceichner@cbs.mpg.de

## EBC Consortium

### Data Analysis and Writing

**Cornelius Eichner[1], Michael Paquette[1], Christian Bock[3], Tobias Gräßle[6], Carsten Jäger[7,8], Evgeniya Kirilina[7,9], Ilona Lipp[7], Markus Morawski[8], Nikolaus Weiskopf[7,10], Roman M. Wittig[11,12,13], Catherine Crockford[11,12,13], Angela D. Friederici[1] & Alfred Anwander[1]**

### Brain Extraction

**Torsten Møller[14] & Karin Olofsson-Sannö[15]**

[14]Kolmården Wildlife Park, Norrköping, Sweden. [15]National Veterinary Institute, Uppsala, Sweden.

# Methods

## Sample preparation

We acquired postmortem MRI data from the brain of a deceased 47-year-old adult female chimpanzee (*Pan troglodytes verus*) from Kolmården Wildlife Park, Sweden. The chimpanzee was medically euthanized due to an untreatable cervical leiomyoma. The brain was extracted and immersion-fixed with 4% paraformaldehyde in phosphate-buffered saline (PBS) at pH 7.4 within a short postmortem interval of only 4 h. The procedures were in line with the ethical guidelines of primatological research at the Max Planck Institute for Evolutionary Anthropology, Leipzig, which were approved by the ethics committee of the Max Planck Society.

After formalin fixation for 6 months, we removed the superficial blood vessels and washed out the paraformaldehyde in PBS for 3 weeks. For MRI scanning, we placed the brain in a spherical acrylic container filled with perfluoropolyether (PFPE). To prevent potential leaking of PFPE during the acquisition, we vacuum-sealed the container using commercially available synthetic foil packaging. We heavily padded both the brain in the sample container as well as the sample itself with sponges to minimize mechanical coupling between the specimen and the MRI system during data acquisition.

## Histology and tissue quality

The tissue quality of the sample was carefully assessed through histology and immunohistochemistry, electron microscopy and an assessment of diffusion FA (Supplementary Note 1).

For a microscopical and histological assessment of the brain tissue quality, we obtained a $2 \times 2 \times 0.5$ cm$^3$ segment from the left cerebellar hemisphere. The tissue segment was cryoprotected in 30% sucrose in PBS with 0.1% sodium azide. A series of frozen sections of 30 µm was cut and sections were collected in PBS with sodium azide. Histology and immunohistochemistry were performed to visualize cytoarchitecture (Nissl stain), myelin (rat anti-myelin basic protein antibody, Abcam), microglia (rabbit anti-IBA-1 antibody, Fujifilm) and astrocytes (rabbit anti-glial fibrillary acidic protein (GFAP) antibody, DAKO). The primary and secondary antibodies used in these experiments are listed at the end of this section.

The primary antibodies used were:

1. Myelinated fibers - detected protein: myelin basic protein (MBP); antibody: rat anti-MBP; dilution: 1:1,000; source: Abcam; cat. no. AB7349; lot no. GR3375915-1
2. Microglia - detected protein: Ionized calcium binding adaptor 1 (Iba-1); antibody: rabbit anti-Iba-1; dilution: 1:2,000; source: Fujifilm; cat. no. Fujifilm 019-19741; lot no. PTN5930
3. Astroglia - detected protein: GFAP; antibody: rabbit anti-GFAP; dilution: 1:5,000; source: Dako; cat. no. Z0334; lot no. 20059855
4. Aβ plaques - detected protein: amyloid-β 17–24; antibody: mouse anti-Aβ clone 4G8; dilution: 1:500; source: BioLegend; cat. no. 800701; lot no. B286227
5. Aβ plaques - detected protein: pyroglutamated amyloid-β Aβ-pE3; antibody: rabbit anti-pE3-Aβ; dilution: 1:500 source: Synaptic Systems; cat. no. 218003; lot no. 218003/6
6. Neurofibrillary tangles - detected protein: hyperphosphorylated tau, pS202/pT205; antibody: mouse anti-pTau clone AT8; dilution: 1:100; source: Thermo Fisher Scientific; cat. no. MN1020

The secondary antibodies used were:

1. Species: donkey anti-mouse; labeling: biotinylated; source: Dianova; cat. no. 715 065 150; lot no. 144671
2. Species: donkey anti-rabbit; labeling: biotinylated; source: Dianova; cat. no. 711 065 152; lot no. 147049
3. Species: donkey anti-rat; labeling: biotinylated; source: Dianova; cat. no. 712 065 150; lot no. 124180

To assess tissue integrity in more detail, ultrastructural assessment was performed based on electron microscopy data (Supplementary Note 1). A small brain slice from the cerebellum (~$20 \times 20 \times 5$ mm) was refixed in 3% paraformaldehyde and 1% glutaraldehyde in 0.2 M sodium cacodylate buffer at pH 7.4 for 48 h. The refixed plate was sectioned at 70 µm using a sectioning vibratome and small triangles (~$3 \times 5$ mm) were cut out. The small triangles were postfixed in buffered 1% osmium tetroxide for 1 h at room temperature with constant shaking, rinsed in PBS, dehydrated in a graded acetone series with a contrast step of 70% acetone with 2% uranyl acetate and embedded in Durcupan Araldite casting resin. For structural orientation, semithin sections were cut at 0.5-µm thickness and stained with toluidine blue. Ultrathin sections (50 nm) were cut on a Reichert Ultramicrotome II. Sections were imaged with a LEO M 912 Omega TEM (Zeiss) at 80 kV. Digital micrographs were acquired with a dual-speed 2K-on-axis CCD camera based on a YAG scintillator and processed with the analytical EM software Image SP (TRS-Tröndle).

## MRI data acquisition

**High-resolution diffusion MRI and anatomical MRI data acquisition.** We acquired whole-brain dMRI data on a preclinical Bruker Biospec 94/30 MRI system at 9.4T (Paravision 6.0.1), using a $G_{max} = 300$ mT/m gradient system and a 154-mm transmit-receive quadrature radiofrequency coil (Bruker BioSpin). The brain was placed in left–right orientation in the center of the system for optimal coil sensitivity. We acquired dMRI data using a segmented 3D EPI spin-echo sequence. The sequence used two adiabatic refocusing pulses[43] to minimize the impact of B1+ inhomogeneity across the sample. We acquired data with an isotropic resolution of 500 µm using the following parameters: TR = 1,000 ms, TE = 58.9 ms, matrix size [r, read × p, phase × s, slice] = 240 × 192 × 144, no partial Fourier, no parallel acceleration, EPI segmentation factor of 32 and EPI-Readout-BW of 400 kHz. Diffusion-weighting was applied with $b = 5,000$ s mm$^{-2}$ in 55 directions, uniformly distributed on a half-sphere and partially flipped to cover the full sphere[44]. Before measurement, we acquired ten diffusion-weighted volumes (almost 13 h) as dummy scans to achieve a constant steady-state temperature in the sample throughout the scanning session. We acquired three interspersed $b = 0$ images without diffusion-weighting for field-drift correction. An additional $b = 0$ volume was acquired with reversed-phase encoding direction to correct off-resonance EPI distortions. Additionally, we recorded a noise map with matching EPI parameters to characterize the noise statistics of the dMRI data. The total dMRI acquisition time was approximately 90 h.

We acquired anatomical 3D FLASH MRI data with identical image dimensions and at the same image resolution (500 µm isotropic) as the dMRI data: TR = 50 ms, TE = 9 ms, matrix size [r × p × s] = 240 × 192 × 144, no partial Fourier, no parallel acceleration, BW = 20 kHz. To generate different contrasts, data with multiple flip angles were acquired with $\vartheta$ = [5, 12.5, 25, 50, 80]°. We obtained an ultra-high-resolution FLASH MR microscopy dataset at 150 µm isotropic resolution using the following parameters: TR = 50 ms, TE = 9 ms, matrix size [r × p × s] = 800 × 640 × 480, $\vartheta$ = 27°, no partial Fourier, no parallel acceleration and BW = 20 kHz. The MR microscopy FLASH acquisition took about 5.5 h.

We repeated both the dMRI and FLASH data acquisitions (at 500 µm) 2 weeks after the initial measurements for a test–retest evaluation.

## MRI data processing

**Diffusion MRI data processing.** Diffusion MRI preprocessing entailed the following seven steps: (1) signal debiasing utilizing the noise s.d., σ, and the effective number of coils, estimated from the noise map[45]; (2) MP-PCA denoising (MRtrix v.3.0.2); (3) 3D volumetric Gibbs ringing correction using sub-voxel shift (MRtrix v.3.0.2); (4) field-drift correction using linear interpolation between the non-diffusion-weighted data; (5) correction of eddy currents from diffusion-weighting and

off-resonance EPI distortions (FSL v.6.0, FSL eddy for CUDA v.8.0, ANTS v.2.3.5)[46]; (6) fitting of a DTI model to the dMRI data to generate maps of FA, radial diffusivity (RD), mean diffusivity (MD) and main fiber orientation (FSL v.6.0); and (7) normalization of the dMRI data and σ map using the mean non-diffusion-weighted $b = 0$ volume. This step also included the calculation of a mean diffusion-weighted dataset, averaged across all diffusion directions for visualization purposes and the test–retest analysis (Python v.3.8).

**Local spherical deconvolution.** We computed fODF using LSD for the whole brain (Python v.3.8). This involved the following steps: (1) the diffusion signal was transformed into a dODF using the constant solid angle (CSA) operation[47]. (2) dODFs were transformed into fODF candidates using the sharpening deconvolution transform[48]. LSD does not assume previous knowledge of the true underlying sharpening ratio. The rotational symmetric kernels of different diffusion anisotropy were automatically selected from a series of predefined ratios (ranging from 1.1 to 6.0), which encode the ratio of diffusivity between the main and the secondary axis of the kernel. The applied ratios represent distinct single-fiber response functions per voxel. (3) The resulting fODF candidates were used to estimate the measured dMRI signal using a multi-tensor diffusion (MTD) model[49]. For this, each candidate's fiber orientations and corresponding peak amplitudes were extracted (relative peak threshold 0.25). The peak amplitudes were used as volume fractions in the MTD model. MTD tensor eigenvalues were uniquely defined by the applied fODF deconvolution ratio and the constraint of the spherical mean of the diffusion-weighted signal[50]. (4) The AIC was used to select the optimal deconvolution ratio for each voxel.

$$AIC = 2k - 2\ln(L)$$

The number of model parameters, $k$, is given by NuFO (each orientation corresponds to three MTD parameters). The model likelihood, $L$, was calculated assuming Gaussian noise with s.d. $\sigma$ (noise map) between the model's data prediction $\hat{S}(\vec{g})$ and the observed data $S(\vec{g})$ for each gradient direction $\vec{g}$.

$$\tilde{L}(\vec{g}) = \frac{1}{\sigma\sqrt{2\pi}} \exp\left[-\frac{1}{2}\left(\frac{\hat{S}(\vec{g}) - S(\vec{g})}{\sigma}\right)^2\right]$$

The directional errors are independent. Hence, the total likelihood across directions was computed as the product of the individual directional likelihoods.

$$L = \prod_{\vec{g}} \tilde{L}(\vec{g})$$

The optimal deconvolution ratio with the lowest AIC in its corresponding MTD was selected for each voxel. For increased spatial consistency, the AIC maps were smoothed using a moderate Gaussian kernel ($\sigma_{smooth} = 0.5$ voxels) before model selection.

To compare the quality of the LSD estimation with conventional methods, fODFs were computed using the CSD[51] algorithm in MRtrix (v.3.0.3). To evaluate LSD and CSD fODF reconstructions, a NuFO map[52] was computed for both methods. LSD and CSD reconstructions were compared in different crossing fiber regions of the brain.

### Test–retest evaluation

We assessed the stability of the dMRI measurements and the LSD fODF reconstructions based on a test–retest evaluation, using a separately acquired dMRI dataset. The retest data were warped to the test data in a one-step correction approach[46], enabling voxel-wise analysis of both datasets without loss in image resolution from multiple interpolations.

Average voxel-wise coefficients of variation were computed to assess the reproducibility between test and retest data with respect to the normalized averaged diffusion signal, FA and the LSD deconvolution ratios.

For a voxel-wise estimation of the stability of estimated fiber orientations, the angles between the reconstructed fiber orientations from the test and retest data were computed. In a first assessment, the angles were calculated for the principal fiber orientation based on DTI. For LSD, these angles were extracted for the reconstructed primary and secondary fiber orientations in WM voxels with two fiber orientations.

**Diffusion MRI tractography.** Whole-brain deterministic streamline fiber tractography was computed based on the LSD fODFs using the MRtrix software tckgen (v.3.0.3, one seed in each voxel, fourth-order Runge–Kutta integration, step size of 0.125 mm, angular threshold of 60° and relative threshold of 0.25). Tractography and seeding were constrained to the segmented brain WM and subcortical regions.

**Track-density images.** We computed a track-density image (TDI) using probabilistic tractography with the following parameters: iFOD2 algorithm, 27 seeds in each voxel, step size of 0.1 mm and relative threshold of 0.1. A dense tractography (larger number of seeds per voxel and smaller step size) was used to generate TDI[53] data with an isotropic resolution of 75 μm. The larger number of generated streamlines in the dense tractography is necessary to compute TDI data of sufficient quality[53]. TDI data features a substantially higher resolution than its original input data and is thus able to resolve finer anatomical details[53]. We used the TDI data to annotate WM regions with anatomical labels[32].

**WM fascicle segmentation.** Principal fiber tracts were segmented and visualized based on the whole-brain tractography using BrainGL (https://github.com/rschurade/braingl). The segmentation of the different fascicles was based on guidelines for humans[31], chimpanzees[12] and macaques[34]. A detailed description of this procedure for each fascicle can be found in Supplementary Note 6.

**Segmentation of WM and subcortical regions.** Brain tissue was segmented using a semi-automated process based on multimodal MRI data. First, a fuzzy c-means algorithm was utilized to obtain clusters from different contrasts (all FLASH contrasts and dMRI b0, FA, MD and RD at 500 μm). The resulting cluster probability maps were then assigned to either WM, gray matter or extra brain space, averaged, median filtered in a $3 \times 3 \times 3$-voxel kernel and binarized (threshold of 0.5). The resulting WM map was then refined manually, based on the multimodal MRI data (ITK-Snap v.3.8.0). In this step, the multimodal MRI data were used to segment the following subcortical structures in both hemispheres: cau, claustrum (clau), globus pallidus (gp), inferior colliculus (ico), lgn, nucleus accumbens (na), pt, red nucleus (rn), substantia nigra (sn), superior colliculus (sc) and the th. The segmentation of these subcortical regions was based on anatomical landmarks. As no chimpanzee brain atlas exists, a human subcortical atlas was chosen for anatomical guidance[54]. Manual segmentations were performed by C.E., A.A. and H.G.

### Assessment and validation of LSD

We undertook a more detailed investigation of the accuracy and performance of the LSD based on (1) ground-truth in silico simulation experiments as well as (2) human postmortem dMRI measurements and histology.

To this end, we first investigated the influence of the measured number of diffusion directions on the accuracy of the LSD. We then examined the overall reconstruction accuracy of LSD and compared it to CSD using Monte Carlo simulations for a wide range of fiber configurations and SNR values.

We then compared the reconstruction accuracy of LSD and CSD based on known anatomy using high-resolution human dMRI measurements. Finally, LSD fiber reconstructions were compared to fiber reconstructions from 2D myelin histology.

**Numerical simulation experiments.** *General simulation setup.* The general process for the simulations involved (1) generating fODFs, (2) generating diffusion kernels, (3) convolving and (4) sampling them to obtain ground truth signals and finally (5) adding noise. The synthetic data were then (6) processed with LSD/CSD, and the reconstructed fODFs were (7) evaluated with various metrics. The synthetic fiber models and dMRI signals were generated using DIPY (https://dipy.org/).

For fODF generation, the individual fODF peaks were generated by sampling the probability density function of a symmetrized Von-Mises Fisher distribution of concentration parameter $\kappa$ and mean orientation $\mu$. The spherical function of multiple such peaks of different orientations was then averaged to obtain the final fODFs in the case of crossing.

The diffusion kernels were modeled as axisymmetric tensors, parametrized by the MD and the diffusivity ratio R (parallel diffusivity over perpendicular diffusivity).

The convolution of the fODFs and tensor kernels was discretely approximated by rotating the tensor over a uniform spherical grid of 724 points.

For diffusion sampling, to generate the diffusion signal, each of the rotated tensors was sampled using the desired $b$ vectors and $b$ values. The resulting signal was averaged using the fODF values as weights. For a single $b$ value shell, the $b$ vectors were typically generated from a uniformly distributed set of $n$ points on the hemisphere using the electrostatic repulsion method[44].

For synthetic noise, the noiseless signals were typically corrupted by adding Gaussian noise (following a distribution with mean 0 and s.d. $\sigma$). Our synthetic signals were normalized such that the signal is 1 at $b = 0$, so we define the SNR as $1/\sigma$. This definition of SNR represents the SNR 'at b0' for each direction.

For fODF reconstruction, the LSD reconstruction followed the pipeline described in the main manuscript. For the simulations, we employed a maximum sh order of 8, deconvolution tensor kernel ratios ranging [1.1, 10], a relative peak extraction threshold of 25% and a minimum peak separation angle of 25 degrees, unless stated otherwise. CSD fODFs were reconstructed in MRtrix.

For evaluation metrics, to evaluate the quality of the reconstructed fODFs, we first computed two main metrics between the extracted peaks and the ground-truth peaks; the difference in their number (NuFO error) and the average angular error between them. To compute the angular error, we looped over all possible pair matches between the extracted peaks and the ground truth peaks and selected the one that minimizes the error. We then computed a derived error metric from both NuFO and angular error, the success-to-attempt-ratio (STAR), which counts the ratio of 'successes' of a given method or parameter choice. Success is defined as a reconstructed fODF with a NuFO error of 0 and an angular error below 5 degrees.

*Impact of acquired diffusion directions on LSD reconstruction.* To evaluate the impact of the number of diffusion directions on the LSD reconstruction quality, we created two typical crossing fiber geometries, generated the diffusion signal for multiple $b$ vector sampling and compared the resulting angular errors. We generated fODFs with both 60° and 90° crossing angles (concentration parameter $\kappa = 24$) and a diffusion kernel with MD of $1 \times 10^{-3}$ mm$^2$ s$^{-1}$ and $R = 3$. We then used a $b$ value of 1,000 s mm$^{-2}$ and generated sets of $b$ vectors for $n = [28:2:200]$ directions. The signal from the fODF and kernel convolution was generated for each set of $b$ vector/$b$ value. The data were then corrupted with Gaussian noise (1,000 noise replicates) for each $n$, using an appropriate

value of $\sigma$ to match the SNR between sets of $b$ vectors. The $\sigma$ was chosen so that sqrt($n$)/sigma was constant and SNR = 100 for $n = 55$. We reconstructed fODFs using LSD (sh order max 6) from this noisy data for both crossing geometries for all numbers of diffusion directions $n$. Finally, we plotted the mean angular error and s.d. (over the 1,000 noise replicates of each $n$).

*Reconstruction accuracy of LSD compared to CSD.* For the simulation experiment 2peaks-manyK, to evaluate the reconstruction accuracy of the LSD method, we generated a large dataset of two-peak signals spanning the space of fODF geometries and the space of diffusion kernel shapes for many SNR. We refer to this dataset as 2peaks-manyK (two fODFs peaks and many diffusion kernels). For this, we generated two-peak fODFs with crossing angles from 30–90° (linearly distributed) and with concentration parameters $\kappa$ from 8 to 24 (log-linearly distributed), for a total of 65 unique fODF shapes. We generated diffusion kernels with MD from $0.6 \times 10^{-3}$ to $1.2 \times 10^{-3}$ mm$^2$ s$^{-1}$ (linearly distributed) and ratios R from 1.5 to 8 (sqrt-linearly distributed), for a total of 30 unique diffusion kernels. We then generated all combinations of fODFs and diffusion kernels, for a total of 1,950 different voxel geometries, excluding rotations.

The signal from the fODF and kernel convolution was generated for $n = 60$ $b$ vectors with a $b$ value of 1,500 s mm$^{-2}$. The data were then overlaid with Gaussian noise, generating data with an SNR of [10:10:100] (100 replicates each). To remove potential orientation bias, we added a random 3D rotation to each fODF before generating the synthetic dMRI signal.

We reconstructed the fODFs for the 2peaks-manyK dataset ($b = 1,500$) using LSD and CSD (MRtrix software). The LSD reconstruction used sh order max 8. The CSD kernel estimation was performed using the 'dwi2response tournier' method on a specially created companion dataset consisting of the same properties as 2peaks-manyK but with only one peak each. We extracted the fODFs peaks and computed the NuFO and angular error. The data were grouped by SNR and crossing angle for each method. For each bin, we computed the ratio of successes to attempts (STAR metric). Success was defined as a reconstruction with the exact number of peaks as the ground truth and a total angular error of <5°. We present the STAR metrics as percentages. We also show the differences between the two methods as percentage points (a value of 10% in the difference plot means that LSD reconstructed 10% more of the total voxels than CSD for that SNR and crossing angle, not 10% more).

For the simulation dataset 2peaks-oneK, we generated another two-peaks dataset, which covers the same range of fODF geometries as 2peaks-manyK but employs only a single diffusion kernel. The fODF parameters are sampled more finely within the same range to obtain the same total number of configurations. We refer to this dataset as 2peaks-oneK (two fODFs peaks and one diffusion kernel). This microstructural configuration is more consistent with the underlying assumptions of CSD, a single diffusion kernel is used to generate the dMRI signal by convolution.

We generated two-peak fODFs with crossing angles ranging from 30–90° (linearly distributed) and concentration parameter $\kappa$ from 8 to 24 (log-linearly distributed), for a total of 195 unique fODF shapes. We employed a diffusion kernel with MD of $0.9 \times 10^{-3}$ mm$^2$ s$^{-1}$ and ratio R of 4.107, which correspond to the median MD and median R from the manyK dataset. We generated all combinations of fODFs and diffusion kernels for a total of 1,950 different voxel geometries, excluding rotations. The signal from the fODF and kernel convolution was generated for $n = 60$ $b$ vectors and a $b$ value of 1,500 s mm$^{-2}$. The data were then corrupted with Gaussian noise, resulting in an SNR of [10:10:100] (1,000 replicates each). We added a random rotation to each fODF before generating the signal to remove potential orientation bias.

We reconstructed the fODF for the 2peaks-oneK dataset ($b = 1,500$) using LSD and CSD (MRtrix software). The LSD reconstruction used

sh order max 8. The CSD kernel estimation was performed using the iterative 'dwi2response tournier' method on a specially created companion dataset consisting of the same microstructural properties as 2peaks-oneK but with only one peak (fiber direction). We extracted the fODF peaks and computed the NuFO and angular error. The data are grouped by SNR and crossing angle for each method. For each bin, we computed the STAR metric. Success was defined as a reconstruction with the exact number of peaks as the ground truth and a total angular error of <5°. We present the STAR metrics as percentages. We also show the differences between the two methods as percentage points (a value of 10% in the difference plot means that LSD reconstructed 10% more of the total voxels than CSD for that SNR and crossing angle, not 10% more).

**Histological assessment of LSD in human sample.** We performed a validation of the reconstructed LSD fiber directions by postmortem measurements of a human brain slab. These data allowed us to (1) conclude the validity of the LSD reconstructions based on the known human anatomy and (2) validate the fiber direction reconstructions using optical histology.

*Sample description.* A 2-cm coronal human brain slab (89 y, 24 h postmortem interval) was obtained through the University of Leipzig's body donation program. Brains were provided by the Paul Flechsig Institute - Centre of Neuropathology and Brain Research, Medical Faculty, Leipzig University, Leipzig, Germany. The entire procedure of case recruitment, acquisition of the patient's personal data, protocols and the informed consent forms, performing the autopsy and handling the autopsy material were approved by the responsible authorities (approval by GZ 01GI9999-01GI0299; approval no. WF-74/16, approval no. 282-02 and approval no. 205/17-ek).

The neuropathologic examination did not reveal any evidence of neurologic disease. Following the standard Brain Bank procedures, the sample was immersion-fixed in either 3% PFA and 1% glutaraldehyde in PBS (pH 7.4) for at least 6 weeks. Following it was embedded in 1.5% agarose in PBS for cutting a 50-µm slice at the vibratome (Hyrax V50, Carl Zeiss Jena). For immunohistochemical processing, the slice was washed with PBS-Tween 20 three times for 7 min. Next, it was immersed in 2% $H_2O_2$ diluted in 60% methanol for 60 min and washed for 10 min in PBS-Tween 20. Incubation in blocking solution (2% BSA, 0.3% milk powder, 0.5% DNS, 0.1% $NaN_3$ in 0.02% PBS-Tween 20) for 60 min followed.

*Diffusion MRI data acquisition and reconstruction.* The sample was rehydrated and embedded in PFPE for MRI acquisition. Slab dMRI data were acquired on a preclinical Bruker Biospec 94/20 MRI system at 9.4T (Paravision 6.0.1), using a $G_{max}$ = 660 mT/m gradient system and an 86 mm transmit-receive quadrature radiofrequency coil (Bruker BioSpin). Data with an isotropic resolution of 400 µm were acquired using a pulsed gradient diffusion-weighted 2D Spin-Echo sequence using the following parameters:

TR = 3,074.5 ms, TE = 38.9 ms, matrix size [r × p × s] = 215 × 175 × 70, no partial Fourier, no parallel acceleration, nine averages. Diffusion-weighting was applied with $b$ = 10,000 s mm$^{-2}$ in 60 directions, on a half-sphere and partially flipped to equally cover the full sphere. Each repetition contained four uniformly interspersed $b$ = 0 acquisitions without diffusion-weighting.

Diffusion data processing entailed (1) removal of data points during the heating phase, (2) noise characterization and signal debiasing using noise standard deviation and the effective number of coils, (3) MP-PCA denoising, (4) Gibbs ringing correction using sub-voxel shift, (5) linear field-drift correction, (6) correction of eddy currents from diffusion-weighting. Finally, we computed fODFs using both CSD and LSD with sh of 8. The CSD computation in MRtrix employed the iterative 'tournier' method[51] to assess a suitable deconvolution kernel.

LSD was computed using a series of predefined deconvolution ratios from 1.1 to 10, a relative peak intensity threshold of 0.25 and an angular peak separation threshold of 20°. The optimal deconvolution ratio was selected in a 3 × 3 × 3 AIC patch.

*Histology data acquisition and reconstruction.* For imaging of the histological slice, the Zeiss CLSM (LSM 880 Airyscan, Carl Zeiss Jena) equipped with a ×20 objective (NA 0.8, WD 0.55 mm, RI 1.38, Plan-Apochromat Carl Zeiss Jena) was used. Autofluorescence of myelinated fibers[55,56] was excited with an argon laser with 488 nm. The emitted light was collected using three-band pass filters at 505–530 nm. The image consists of 25 × 27 single microscopic images, here referred to as tiles. By deploying the tile scan function of the CLSM, a larger mosaic containing the entire histological slide was acquired. Each tile size was 120 × 120 µm.

### Reporting summary

Further information on research design is available in the Nature Portfolio Reporting Summary linked to this article.

## Data availability

The resource presented here includes (1) dMRI data at 500-µm isotropic resolution, (2) MR microscopy FLASH data at 150-µm isotropic resolution, (3) anatomical FLASH data at 500-µm isotropic resolution, (4) WM fiber pathway reconstructions, (5) ultra-high-resolution TDI data and (6) segmentations of various anatomical brain structures. The raw data and all other features of the resource can be downloaded at https://open-science.cbs.mpdl.mpg.de/ebc/ (https://doi.org/10.17617/3.O5XSI9).

The data volume and 3D-reconstructed fascicles can be viewed online on the open science framework of the Max Planck Institute for Human Cognitive and Brain Sciences at https://openscience.cbs.mpdl.mpg.de/ebc/.

## Code availability

Code and processing routines are publicly available for download at https://github.com/cornelius-eichner/EBC_dMRI_Preprocessing (MPL-2.0 license).

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

## Acknowledgements

This study was funded by the Max Planck Society under the inter-institutional funds of the president of the Max Planck Society for the project 'Hominoid Brain Connectomics' (M.IF.A.XXXX8103). We thank the Evolution of Brain Connectivity (EBC) Consortium members for their ongoing effort to provide high-quality tissue samples. We especially thank S. Unwin from the University of Birmingham, as well as A. Düx and F. Leenderz from the German Robert Koch Institute for their support in creating the brain extraction network. We are grateful to the Kolmården Wildlife Park in Kolmården, Sweden for performing the brain extraction and providing the tissue sample. We thank H. Gerbeth, A. Jauch, K. Reimann and D. Edler von der Planitz for their support during the preparation of the tissue sample. We are grateful to H. Gerbeth and T. Lelke for their support during the MRI acquisitions and segmentation. We thank T. Basse-Lüsebrink from Bruker BioSpin for helpful discussions in setting up the imaging sequences.

## Author contributions

C.E. was responsible for conceptualization, methodology, software, investigation, formal analysis, writing of the original draft, data curation and visualization. M.P. was responsible for methodology, software, investigation, data curation, review and editing. C.M.A. was responsible for methodology, investigation, writing of the original draft, review and editing. C.B. was responsible for resources, investigation, review and editing. E.B. was responsible for resources, investigation, review and editing. T.G. was responsible for methodology and resources. C.J. was responsible for investigation and writing of the original draft. E.K. was responsible for methodology, review and editing. I.L. was responsible for methodology, review and editing. M.M. was responsible for methodology, review and editing. H.R. was responsible for investigation, review and editing. P.W. was responsible for investigation, review and editing. N.W. was responsible for methodology, review and editing and funding acquisition. R.M.W. was responsible for conceptualization, resources, review and editing and funding acquisition. C.C. was responsible for conceptualization, resources, review and editing and funding acquisition. A.D.F. was responsible for conceptualization, review and editing, supervision and funding acquisition. A.A. was responsible for conceptualization, methodology, formal analysis, investigation, data curation, writing of the original draft and visualization.

## Funding

## Competing interests

The authors declare no competing interests.

## Additional information

**Correspondence and requests for materials** should be addressed to Cornelius Eichner.

# Reporting Summary

## Statistics

For all statistical analyses, confirm that the following items are present in the figure legend, table legend, main text, or Methods section.

| n/a | Confirmed | |
|-----|-----------|---|
| ☒ | ☐ | The exact sample size (*n*) for each experimental group/condition, given as a discrete number and unit of measurement |
| ☒ | ☐ | A statement on whether measurements were taken from distinct samples or whether the same sample was measured repeatedly |
| ☒ | ☐ | The statistical test(s) used AND whether they are one- or two-sided *Only common tests should be described solely by name; describe more complex techniques in the Methods section.* |
| ☒ | ☐ | A description of all covariates tested |
| ☒ | ☐ | A description of any assumptions or corrections, such as tests of normality and adjustment for multiple comparisons |
| ☒ | ☐ | A full description of the statistical parameters including central tendency (e.g. means) or other basic estimates (e.g. regression coefficient) AND variation (e.g. standard deviation) or associated estimates of uncertainty (e.g. confidence intervals) |
| ☒ | ☐ | For null hypothesis testing, the test statistic (e.g. *F*, *t*, *r*) with confidence intervals, effect sizes, degrees of freedom and *P* value noted *Give P values as exact values whenever suitable.* |
| ☒ | ☐ | For Bayesian analysis, information on the choice of priors and Markov chain Monte Carlo settings |
| ☒ | ☐ | For hierarchical and complex designs, identification of the appropriate level for tests and full reporting of outcomes |
| ☒ | ☐ | Estimates of effect sizes (e.g. Cohen's *d*, Pearson's *r*), indicating how they were calculated |

*Our web collection on statistics for biologists contains articles on many of the points above.*

## Software and code

Policy information about availability of computer code

| | |
|---|---|
| Data collection | High Resolution Post-mortem MRI data were collected on a 9.4T Bruker Biospec 94/30 System, operated with Paravision version 6.01 Diffusion Pre-Scans were acquired on a 3T Siemens Connectom MRI System. |
| Data analysis | MRI data were processed and analzed using the following software:<br>- ANTs (v2.3.5)<br>- Braingl (v0.20170310)<br>- Bruker2NIfTI (v1.0.20180303)<br>- FSL (v6.0) and<br>- FSL eddy (build 508) for CUDA (v8.0)<br>- ITK-SNAP (v3.8.0)<br>- MRTRIX (v3.0.2)<br>- Python (v3.8) in conjunction with DiPy (v1.3.0), NiBabel (v3.1.1), NumPy (v1.18.5), SciPy (v1.5.0), and ScilPy (v1.1)<br><br>The data analysis pipeline is accessible under https://github.com/cornelius-eichner/EBC_dMRI_Preprocessing |

For manuscripts utilizing custom algorithms or software that are central to the research but not yet described in published literature, software must be made available to editors and reviewers. We strongly encourage code deposition in a community repository (e.g. GitHub). See the Nature Portfolio guidelines for submitting code & software for further information.

## Data

Policy information about availability of data

All manuscripts must include a data availability statement. This statement should provide the following information, where applicable:
- Accession codes, unique identifiers, or web links for publicly available datasets
- A description of any restrictions on data availability
- For clinical datasets or third party data, please ensure that the statement adheres to our policy

The resource presented here includes (i) dMRI data at 500 μm isotropic resolution, (ii) MR-microscopy FLASH data at 150 μm isotropic resolution, (iii) anatomical FLASH data at 500 μm isotropic resolution, (iv) WM fiber pathway reconstructions , (v) ultra-high-resolution TDI data, and (vi) segmentations of various anatomical brain structures. The raw data and all other features of the resource can be downloaded at https://openscience.cbs.mpdl.mpg.de/ebc/.
The data volume and 3D-reconstructed fascicles can be viewed online on the open science framework of the Max Planck Institute for Human Cognitive and Brain Sciences at https://openscience.cbs.mpdl.mpg.de/ebc/.

Code and processing routines are publicly available for download at https://github.com/cornelius-eichner/EBC_dMRI_Preprocessing.

# Field-specific reporting

Please select the one below that is the best fit for your research. If you are not sure, read the appropriate sections before making your selection.

☒ Life sciences      ☐ Behavioural & social sciences      ☐ Ecological, evolutionary & environmental sciences

For a reference copy of the document with all sections, see nature.com/documents/nr-reporting-summary-flat.pdf

# Life sciences study design

All studies must disclose on these points even when the disclosure is negative.

| Sample size | One chimpanzee brain was selected for MRI data acquisition and measured twice with identical parameters. <br><br> One human in-vivo brain dataset was randomly selected from the 7T Human Connectome Project for visual comparison. <br> One human in-vivo brain dataset was randomly selected from the Human Connectome Project for Number of Fiber Orientations comparison. <br> One coronal human brain slab was obtained through the University of Leipzig's body donation program for MRI scanning and validation. <br><br> A sample size estimation was not applicable as this resource entails data from a single subject. |
|---|---|
| Data exclusions | No data were excluded. |
| Replication | To guarantee the stability of the results, an extensive test/retest analysis of the data set was performed. <br><br> The data collection was independently repeated once, two weeks after the initial collection. Data quality was confirmed to be reproducible in this experiment. |
| Randomization | Randomization was not required. The resource contains one brain sample. |
| Blinding | Blinding was not applicable in this study. Samples were analyzed identical and operator-independent procedures. |

# Reporting for specific materials, systems and methods

We require information from authors about some types of materials, experimental systems and methods used in many studies. Here, indicate whether each material, system or method listed is relevant to your study. If you are not sure if a list item applies to your research, read the appropriate section before selecting a response.

## Materials & experimental systems

| n/a | Involved in the study |
|---|---|
| ☐ | ☒ Antibodies |
| ☒ | ☐ Eukaryotic cell lines |
| ☒ | ☐ Palaeontology and archaeology |
| ☐ | ☒ Animals and other organisms |
| ☒ | ☐ Human research participants |
| ☒ | ☐ Clinical data |
| ☒ | ☐ Dual use research of concern |

## Methods

| n/a | Involved in the study |
|---|---|
| ☒ | ☐ ChIP-seq |
| ☒ | ☐ Flow cytometry |
| ☐ | ☒ MRI-based neuroimaging |

# Antibodies

| | |
|---|---|
| Antibodies used | Primary Antobodies:<br>1) Myelinated Fibers - Detected Protein: Myelin basic protein (MBP); Antibody: Rat anti-MBP; Dilution: 1:1000; Source: Abcam; Cat#: AB7349; Lot# GR3375915-1<br>2) Microglia - Detected Protein: Ionized calcium binding adaptor 1 (Iba-1);  Antibody: Rabbit anti-Iba-1;  Dilution: 1:2000; Source: Fujifilm; Cat#: Fujifilm 019-19741; Lot#: PTN5930<br>3) Astroglia - Detected Protein: Glial fibrillary acidic protein (GFAP); Antibody: Rabbit anti-GFAP; Dilution: 1:5000; Source: Dako; Cat#: Z0334; Lot#: 20059855<br>4) Aß plaques - Detected Protein: Amyloid-β 17–24; Antibody: Mouse anti-Aß clone 4G8; Dilution: 1:500; Source: Biolegend; Cat#: 800701; Lot#: B286227<br>5) Aß plaques - Detected Protein: Pyroglutamated amyloid-β Abeta-pE3; Antibody: Rabbit anti-pE3-Aß; Dilution: 1:500 Source: Synaptic Systems; Cat#: 218003; Lot#: 218003/6<br>6) Neurofibrillary Tangles - Detected Protein: hyperphosphorylated tau, pS202/pT205; Antibody: Mouse anti-pTau clone AT8; Dilution: 1:100; Source: Thermo Fisher Scientific; Cat#: MN1020<br><br>Secondary Antibodies:<br>Species: Donkey anti-Mouse; Labeling: biotinylated; Source: Dianova; Cat#: 715 065 150; Lot#: 144671<br>Species: Donkey anti-Rabbit; Labeling: biotinylated;  Source: Dianova; Cat#: 711 065 152; Lot#: 147049<br>Species: Donkey anti-Rat; Labeling: biotinylated; Source: Dianova; Cat#: 712 065 150; Lot#: 124180 |
| Validation | Primary antibodies 1 - 3 have been validated in the following study:<br>Morawski M., Kirilina E., Scherf N., Jäger C., Reimann K., Trampel R., Gavriilidis F., Geyer S., Biedermann B., Arendt T., Weiskopf N. Developing 3D microscopy with CLARITY on human brain tissue: Towards a tool for informing and validating MRI-based histology. NeuroImage 182: 417-428 (2018).<br><br>Primary antibodies 4 - 6 have been validated in the following study:<br>Schober R., Hilbrich I., Jäger C., Holzer M. Senile plaque calcification of the lamina circumvoluta medullaris in Alzheimer's disease. Neuropathology 41(5):366-370 (2021) |

# Animals and other organisms

Policy information about studies involving animals; ARRIVE guidelines recommended for reporting animal research

| | |
|---|---|
| Laboratory animals | No laboratory animals were used in this study. |
| Wild animals | No wild animals were used in this study.<br><br>Data were acquired post-mortem from the brain of a deceased 47-year-old adult female chimpanzee (pan troglodytes verus) from Kolmården Wildlife Park, Sweden. The zoo chimpanzee was medically euthanized due to an untreatable cervical leiomyoma |
| Field-collected samples | No field collected samples were used in this study. |
| Ethics oversight | The procedures were in line with the ethical guidelines of primatological research at the Max Planck Institute for Evolutionary Anthropology, Leipzig, which were approved by the ethics committee of the Max Planck Society. |

Note that full information on the approval of the study protocol must also be provided in the manuscript.

# Magnetic resonance imaging

## Experimental design

| | |
|---|---|
| Design type | Structural Imaging |
| Design specifications | Not applicable, as no functional MRI data were collected. |
| Behavioral performance measures | Not applicable, as MRI data were post mortem. |

## Acquisition

| | |
|---|---|
| Imaging type(s) | Diffusion and Structural MRI |
| Field strength | 9.4T |
| Sequence & imaging parameters | Chimpanzee MRI data were acquired on a preclinical Bruker Biospec 94/30 MRI system at 9.4T (Paravision 6.0.1), using a 300 mT/m gradient system and a 154 mm transmit-receive quadrature RF coil (Bruker BioSpin, Ettlingen, Germany).<br><br>Diffusion MRI: |

Segmented 3D EPI sequence with double adiabatic refocusing, TR = 1000 ms, TE = 58.9 ms, matrix size [r × p × s] = 240 × 192 × 144, no Partial Fourier, no parallel acceleration, EPI segmentation factor = 32 EPI-BW = 400 kHz. Diffusion-weighting was applied with b = 5.000 s/mm2 in 55 directions, uniformly distributed on a full sphere. Before the acquisition, ten diffusion-weighted volumes were acquired as dummy scans to achieve a constant steady-state temperature in the sample. Three interspersed b = 0 images without diffusion-weighting were acquired for field-drift correction. An additional b = 0 volume was acquired with reversed-phase encoding direction to correct off-resonance EPI distortions. A noise map with matching EPI parameters was additionally recorded to characterize the noise statistics of the dMRI data.

Anatomical Structural MRI:
Anatomical 3D FLASH MRI data were acquired using identical image dimensions and image resolution (500 μm isotropic) as the dMRI data: TR = 50 ms, TE = 9 ms, Matrix Size [r × p × s] = 240 × 192 × 144, no Partial Fourier, no parallel acceleration, BW = 20 kHz. To generate different contrasts, data with multiple flip angles were acquired with ? = [5, 12.5, 25, 50, 80]°. In addition, an ultra-high-resolution FLASH MR-microscopy dataset was acquired at 150 μm resolution using the following parameters: TR = 50 ms, TE = 9 ms, Matrix Size [r × p × s] = 800 × 640 × 480, ? = 27°, BW = 20 kHz.

The 500 μm dMRI and FLASH data acquisitions were repeated two weeks after the initial measurements for a test-retest evaluation.

Human MRI data were acquired on a preclinical Bruker Biospec 94/20 MRI system at 9.4T (Paravision 6.0.1), using a 660 mT/m gradient system and a 86 mm transmit-receive quadrature RF coil (Bruker BioSpin, Ettlingen, Germany).

Diffusion MRI:
Data with an isotropic resolution of 400 μm were acquired using a pulsed gradient diffusion weighted 2D Spin-Echo sequence using the following parameters: TR = 3074.5 ms, TE = 38.9 ms, matrix size [r × p × s] = 215 × 175 × 7, no Partial Fourier, no parallel acceleration, 9 averages. Diffusion-weighting was applied with b = 10.000 s/mm2 in 60 directions, on a half-sphere, and partially flipped to equally cover the full sphere. Each repetition contained 4 uniformly interspersed b=0 acquisitions without diffusion weighting.

| Area of acquisition | whole brain (chimpanzee), coronal slab (human) |

Diffusion MRI ☒ Used ☐ Not used

| Parameters | Chimpanzee: Diffusion MRI with 55 directions, single shell at b = 5.000 s/mm2. No cardiac gating<br>Human Diffusion MRI with 60 directions, single shell at b = 10.000 s/mm2. No cardiac gating |

## Preprocessing

| Preprocessing software | Diffusion MRI preprocessing of the chimpanzee dataset entailed the following steps:<br>(i) Signal debiasing using Python (v3.8) in conjunction with autodmri (v0.2.5), DiPy (v1.3.0), NiBabel (v3.1.1), NumPy (v1.18.5), and SciPy (v1.5.0)<br>(ii) MP-PCA denoising using MRtrix (v3.0.2) dwidenoise<br>(iii) Volumetric Gibbs-ringing correction using MRtrix (v3.0.2) with custom built MRdegibbs3D<br>(iv) Field-drift correction using Python (v3.8) in conjunction with NiBabel (v3.1.1), NumPy (v1.18.5), and SciPy (v1.5.0)<br>(v) Eddy current and distortion correction using FSL (v6.0) eddy (build 508) for CUDA (v8.0)<br>(vi) DTI model fit using FSL (v6.0) dtifit<br>(vii) dMRI data normalization with non-diffusion-weighted volumes using Python (v3.8) in conjunction with NiBabel (v3.1.1), NumPy (v1.18.5)<br>(viii) Local Spherical Deconvolution using Python (v3.8) in conjunction DiPy (v1.3.0), NiBabel (v3.1.1), NumPy (v1.18.5), and SciPy (v1.5.0)<br>(ix) WM segmentaion using ITK-SNAP (v3.8.0)<br>(x) MR Tractography using MRTRIX (v3.0.2) tckgen |
| Normalization | The single subject data were not registered to a group template |
| Normalization template | N/A |
| Noise and artifact removal | Image noise was removed using MP PCA denoising, implemented in MRtrix (v3.0.2) dwidenoise |
| Volume censoring | N/A |

## Statistical modeling & inference

| Model type and settings | N/A |
| Effect(s) tested | N/A |

Specify type of analysis: ☒ Whole brain ☐ ROI-based ☐ Both

| Statistic type for inference<br>(See Eklund et al. 2016) | N/A |
| Correction | N/A |

## Models & analysis

| n/a | Involved in the study |
|-----|----------------------|
| ☒ ☐ | Functional and/or effective connectivity |
| ☒ ☐ | Graph analysis |
| ☒ ☐ | Multivariate modeling or predictive analysis |

