## [Peer Review File · Nature Methods]

Peer Review Information

Manuscript Title: Detailed Mapping of the Complex Fiber Structure and White Matter Pathways of the Chimpanzee Brain

Corresponding author name(s): Cornelius Eichner

Editorial Notes: None

Reviewer Comments & Decisions:

Decision Letter, initial version:

Dear Dr Eichner,

Let me first apologize for the delays in the review process. Your Resource entitled "A Resource for Detailed Mapping of the Complex Fiber Structure and White Matter Pathways of the Chimpanzee Brain" has now been seen by two reviewers, whose comments are attached. While they find your work of potential interest, they have raised serious concerns which in our view are sufficiently important that they preclude publication of the work in Nature Methods, at least in its present form.

As you will see, the reviewers raise concerns about the advance of your resource over published work, about potential degradation in the sample and that the data is based on one animal only.

Should further experimental data allow you to fully address these criticisms we would be willing to look at a revised manuscript (unless, of course, something similar has by then been accepted at Nature Methods or appeared elsewhere). This includes submission or publication of a portion of this work somewhere else. We hope you understand that until we have read the revised paper in its entirety we cannot promise that it will be sent back for peer-review.

If you are interested in revising this manuscript for submission to Nature Methods in the future, please contact me to discuss your appeal before making any revisions. Otherwise, we hope that you find the reviewers' comments helpful when preparing your paper for submission elsewhere.

I would also like to renew my apologies for the delays.

Best regards,
Nina

Nina Vogt, PhD
Senior Editor

Nature Methods

Reviewers' Comments:

Reviewer #1:

Remarks to the Author:

In this study, Eichner and colleagues provide the resource of the high-resolution ex vivo dMRI dataset acquired from a chimpanzee brain. They described a method to acquire high spatial resolution dMRI dataset, a method to model fiber orientation distribution, and provided visualization of white matter tracts using both diffusion-encoded color maps and tractography.

Overall, this is an essential resource considering an ongoing interest on comparative diffusion MRI and difficulty to access to the chimpanzee brain. While I am supportive for this work, I have some major comments on a current version of the manuscript.

Major comments:

(1) Authors repeatedly used a term "resolution" thorough the manuscript. However, I think that a true spatial resolution is not equal to the voxel size considering a possible existence of signal blurring. I would recommend using a term "voxel size", rather than "resolution", since it is practically difficult to perform an accurate measurement on an effective spatial resolution.

(2) I am not fully convinced about the advantage of LSD-fODF over CSD, because there is no ground truth on chimpanzee fiber pathway. For this reason, it is difficult to establish which voxelwise model works better or not. Although authors performed test-retest analysis on LSD-fODF, did you perform the same analysis on CSD? One possible idea to strengthen author's conclusion is having a look at brain regions in which a pattern of fiber crossing is relatively well known, such as the optic chiasm.

(3) Authors measured 58 uniformly distributed diffusion direction. This is not particularly high angular resolution data compared with available human dataset (e.g., the HCP Young Adult Dataset). I assume that this is because acquisition time is very long because authors used readout segmentation; however, please make explicit statements on why authors made this choice, rather than other choices (such as increasing diffusion direction or performing multi-shell measurements). In addition, considering a limited angular resolution, how much authors are confident about an ability of LSD-fODF to resolve crossing fibers?

(4) I do not fully understand a comparison with the Bryant dataset in Supplementary Information. Specifically, I could not follow from where differences in tract trajectory are coming from. While I am convinced that a present study provides higher quality data, Bryant study may use different ROI definition and tractography algorithm. I suspect that a difference in tract trajectory between studies can be mostly explained by differences in ROI definition. How much authors used similar or different placement of ROIs? If authors used similar ROIs as Bryant paper did, how much differences in tract trajectory remain? It is difficult to establish what is a right ROI selection due to a limited number of previous studies on chimpanzee brains. Therefore I think that it is difficult to prove that author's ROI selections is superior to those by Bryant et al.

(5) The National Chimpanzee Brain Resource by Chet Sherwood, William D. Hopkins and Todd M. Preuss (<https://www.chimpanzeebrain.org>) should be mentioned as an existing database of chimpanzee dMRI dataset. The advantage of the present dataset compared with this existing resource should be discussed. Authors used a term "Chimpanzee Resource" in Supplementary material, but this can be potentially puzzling since a term is too similar to the National Chimpanzee Brain Resource.

Specific comments:

(6) [Introduction, third paragraph] "Primate brain evolution is characterized not only by a rapid increase in brain size and complexity but also but a marked increase in the proportion of white matter" Authors may consider to cite a review paper from Thiebaut de Schotten et al. *Cortex* (2019; 118, 188-202) which provides an overview of comparative dMRI and also refers chimpanzee studies.

(7) [Methods, "Diffusion MRI Tractography"] Authors stated that they used the LSD-fODF and MRTrax for tractography but is this really implemented in MRTrax or did author implement this processed as a custom code?

(8) SLF I reconstruction provided in a supplementary information show two explicit branches; is this an expected pattern or due to a limited ability of resolving crossing fibers?

Reviewer #2:

Remarks to the Author:

Eichner et al present a high-resolution post-mortem MRI data set of one Chimpanzee monkey brain i.e., great apes that evolution-wise is believed very close to the human brain organization. The authors have collected a high-resolution whole-brain diffusion MRI data set in isotropic 0.5 mm voxel which is a bit higher than the isotropic 0.6 mm image resolution presented by others such as Roumazeilles et al (2020, *Plos Biology*) and the 1.8 mm by Bryant et al (2020, *Plos Biology*) though collected in vivo in a larger cohort. The authors did optimize their acquisition parameters (the b-value) for ex vivo dMRI to secure the best contrast to separate crossing fibers. The data set contains a single b-value acquired in 58 directions. The diffusion tensor model (single fiber) and the conventional constrained spherical deconvolution (CSD) multi-fiber model were fitted to the dMRI data set. Also, the authors present the multi-fiber (Local Spherical Deconvolution) LSD model that appears to model fewer apparently spurious fiber directions per voxel than the CSD. The CSD multi-fiber model is an established model with applications in both in vivo and ex vivo data sets. It is unclear why the CSD model when applied to the presented dMRI data set seems to produce spurious fiber directions. Maybe it is related to a kind of degradation of the tissue quality compared to in vivo due to the use of the immersion fixation procedure after a post-mortem interval of four hours. It is well-described that myelin layers normally tightly wrapped around axons are very sensitive to autolytic effects and become loose (Dyrby et al. 2018, *Neuroimage*). Such myelin loosening effect could be a source of the spurious fiber direction seen in CSD while the presented LSD multi-fiber model may suppress the autolytic effect. Indeed, the authors did check the tissue quality using different staining using light microscopy, but the technique lacks image resolution to the integrity of myelin layers. Validation of the LSD model using for example a structure-tensor model applied to histology may have shed light on the question. Nevertheless, the anatomical information of white matter pathways revealed with the color-coded diffusion tensor model and tract density maps indicates a high quality of the dMRI data set. Both the major and finer white matter pathways are detected. The virtually traced pathways using tractography appear dense and show fine tract details likely due to a relatively high image resolution compared to the circumference of the tracts - i.e., a low partial volume effect.

I appreciate the appealingly high quality of the data set and the sensitivity to fine anatomical details emphasized. It is good for the community to share such data sets. However, the scientific impact of providing a single post-mortem data set with only a single standard b-value shell for tractography application is not clear compared to already available in vivo and ex vivo data sets.

1) To reduce the post-mortem interval and the risk of introducing the degrading autolytic effects it is

unclear why the authors did not perform perfusion fixation on the chimpanzee.

2) It is known that chemical fixation reduces the diffusivity in the tissue and the b-value need to be increased accordingly as the authors also did. If the post-mortem tissue quality is high the FA value should be like in vivo values (Sun et al., 2005, MRM). The authors do show a FA map (Fig S3,B) and show a color bar hard to interpret. For the reader to appreciate the data quality the authors should quantify if the FA values in CC are like in vivo values in the litterateur. I would expect the FA values to be like or higher than in vivo (also to in vivo humans) due to the high image resolution hence a lower partial volume effect in CC.

3) The authors should better explain the need to present a new multi-fiber LSD model compared with the standard CSD. It is not clear why a new yet not validated fiber model is to be introduced when the focus is to present a high-quality dMRI data set expected to work with standard fiber models. The fiber model comparison results presented in figure 2 are not convincing. That said, I think the LSD model approach is appealing.

4) Line 241: Chimpanzees are monkeys.

5) Line 255: "Laminae guide cc axons while growing..." How can a laminae structure act as a guide?

6). Line 260: The authors say that columnar organization in CC is visible with the diffusion tensor model as well as with the high-resolution FLASH MR data. I do not see any such organization in FLASH MRI data when inspecting figure 3.

7) Liner 264 - 271: I do not follow the callosal ratio results presented. Where do the authors show the ratios?

8) Line 272: "... importance to the evolution of brain ...". Are the ratios corrected for brain size differences?

9) Line 285: Please rephrase the sentence.

10) Figure 5: Why did the authors not perform Klingner dissection on the brain instead of showing Klingner dissection of a human brain?

Methods:

11) The authors say that the signal-to-noise ratio is high, but it seems that the SNR is not reported.

12) How many times was the diffusion MRI data set repeated and how were the data sets averaged to increase the SNR?

13) Line 485: What was the averaged DWI data set used for?

14) When performing the retest of the dMRI data set the authors used non-linear registration to align the two data sets and argue it is due to gradient non-linearities of the gradient system. However, a preclinical MRI scanner was used where the gradient non-linearities usually are not a problem. Could the non-linear differences observed be due to mechanical instabilities of the postmortem brain placed in the container? How did the authors ensure the brain is mechanically stable during scanning, storing,

and rescanning?

15) The authors mention the challenges of temperature drifts of the brain tissue due to hardware heating of the gradient hardware. It is not clear how the authors practically ensured to reduce such potential temperature shifts. Did they use a constant air-conditioned flow around the tissue? Maybe minor temperature drifts do not matter?

Author Rebuttal to Initial comments

Reviewer 1

Remarks to the Author

In this study, Eichner and colleagues provide the resource of the high-resolution ex vivo dMRI dataset acquired from a chimpanzee brain. They described a method to acquire high spatial resolution dMRI dataset, a method to model fiber orientation distribution, and provided visualization of white matter tracts using both diffusion-encoded color maps and tractography.

Overall, this is an essential resource considering an ongoing interest on comparative diffusion MRI and difficulty to access to the chimpanzee brain. While I am supportive for this work, I have some major comments on a current version of the manuscript.

Response: We thank the reviewer for his general support. We performed further experiments and manuscript modifications to address the remaining concerns.

Major Comments

1. Authors repeatedly used a term “resolution” thorough the manuscript. However, I think that a true spatial resolution is not equal to the voxel size considering a possible existence of signal blurring. I would recommend using a term “voxel size”, rather than “resolution”, since it is practically difficult to perform an accurate measurement on an effective spatial resolution.

Response: The reviewer is correct in pointing out the difference between resolution and voxel size. We changed the manuscript accordingly. However, we would like to emphasize that the optimized segmented 3D EPI sequence used allowed us to reduce signal blurring and effectively increase resolution compared to previously acquired datasets.

2. I am not fully convinced about the advantage of LSD-fODF over CSD, because there is no ground truth on chimpanzee fiber pathway. For this reason, it is difficult to establish which voxelwise model works better or not. Although authors performed test-retest analysis on LSD-fODF, did you perform the same analysis on CSD? One possible idea to strengthen author’s conclusion is having a look at brain regions in which a pattern of fiber crossing is relatively well known, such as the optic chiasm.

Response: We thank the reviewer for the specific interest in our LSD fiber reconstruction method. We agree with the reviewer that a ground truth measurement is required to evaluate the accuracy of our introduced reconstruction method. A test-retest analysis of the CSD reconstruction would not support our analysis, as we argue for a reconstruction bias (i.e., too many numbers of fiber orientations) which might be reproducible, yet incorrect, between experiments.

A fully quantitative and comprehensive comparison between both reconstruction methods may only be performed through comparison against a ground truth, which is available *in-silico*. In addition, *post-mortem* measurements in a known anatomy may provide additional confidence in the observed reconstruction differences. To address the reviewer's point, we have performed an extensive validation of LSD as well as a comparison with CSD. This comparison includes (i) extensive numerical simulations against *in-silico* ground truth, (ii) a comparison between LSD and CSD in a high resolution dMRI dataset of a *post-mortem* human (with known anatomy), as suggested by the reviewer, and (iii) a comparison between LSD and *post-mortem* histology.

We find that LSD substantially outperformed CSD in reconstruction of crossing fibers – especially for typically challenging sharp crossings. In the additional human *post-mortem* comparison, we observed that CSD introduced a large number of spurious directions, which were removed using LSD.

Our validation experiments demonstrate the high accuracy of LSD fiber reconstruction compared to CSD and will help build confidence in the model within the research community. The results of this validation are included in the Supplementary Information. As the additional analyses substantially strengthened the paper, we would like to thank the reviewer for this suggestion.

Manuscript Changes:

Results:

[...] The SI provides a thorough validation of LSD reconstruction accuracy against the ground truth from numerical simulations, known human anatomy, as well as myelin histology. This validation also entails a detailed comparison and benchmark of LSD against CSD, as well as an analysis of LSD reconstruction stability against the acquired number of diffusion directions. [...]

Discussion:

[...] Extensive computational and histologic validation, as well as test-retest analysis, demonstrated the superior accuracy and stability of the developed LSD algorithm (See SI). [...]

Supplementary Information:

4. Assessment and Validation of Local Spherical Deconvolution

In the following, we describe a more detailed investigation of the accuracy and performance of the LSD based on (i) ground truth *in-silico* simulation experiments as well as (ii) human *post-mortem* dMRI measurements and histology.

[...]

We then compared the reconstruction accuracy of LSD and CSD based on known anatomy using high-resolution human dMRI measurements. Finally, LSD fiber reconstructions were compared to fiber reconstructions from 2D myelin histology.

4.1 Numerical Simulation Experiments

General Simulation Setup

The general process for the employed simulations involved 1) generating fODFs, 2) generating diffusion kernels, 3) convolving and 4) sampling them to obtain ground truth signals, and finally 5) adding noise. The synthetic data were then 6) processed with LSD, and the reconstructed fODFs are 7) evaluated with various metrics. The synthetic fiber models and dMRI signals were generated using DIPY (<https://dipy.org/>).

fODF generation: The individual fODF peaks were generated by sampling the probability density function of a symmetrized Von-Mises Fisher distribution of concentration parameter κ and mean orientation μ . The spherical function of multiple such peaks of different orientations was then averaged to obtain the final fODFs in the case of crossing.

Diffusion Kernels: The diffusion kernels were modeled as axisymmetric tensors, parametrized by the mean diffusivity MD, and the diffusivity ratio R (parallel diffusivity over perpendicular diffusivity).

Kernel Convolution: The convolution of the fODFs and tensor kernels was discretely approximated by rotating the tensor over a uniform spherical grid of 724 points.

Diffusion Sampling: To generate the diffusion signal, each of the rotated tensors was sampled using the desired b-vectors and b-values. The resulting signal was averaged using the fODF values as weights. For a single b-value shell, the b-vectors were typically generated from a uniformly distributed set of N points on the hemisphere using the electrostatic repulsion method (Jones et al., 1999).

Synthetic Noise: The noiseless signals were typically corrupted by adding Gaussian noise (following a distribution with mean 0 and standard deviation σ). Our synthetic signals were normalized such that the signal is 1 at b-value=0, so we define the SNR as $1/\sigma$. This definition of SNR represents the SNR "at b0" for each direction.

fODF Reconstruction: The LSD reconstruction followed the pipeline described in the main manuscript. For the simulations, we employed a maximum sh order of 8, deconvolution tensor kernel ratios ranging in [1.1, 10], a relative peak extraction threshold of 25%, and a minimum peak separation angle of 25 degrees – unless stated otherwise. CSD fODFs were reconstructed in MRtrix.

Evaluation Metrics: To evaluate the quality of the reconstructed fODFs, we first computed two main metrics between the extracted peaks and the ground truth peaks; the difference in their number (NuFO error) and the average angular error between them. To compute the angular error, we looped over all possible pair matches between the extracted peaks and the ground truth peaks and selected the one that minimizes the error. We then computed a derived error metric from both NuFO and angular error - the Success-To-Attempt-Ratio (STAR), which counts the ratio of "successes" of a given method or parameter choice. Success is defined as a reconstructed fODF with a NuFO error of 0 and an angular error below 5 degrees.

Reconstruction Accuracy of LSD Compared to CSD

Simulation Experiment 2peaks-manyK: To evaluate the reconstruction accuracy of the LSD method, we generated a large dataset of 2-peak signals spanning the space of fODF geometries and the space of diffusion kernel shapes for many SNR. We refer to this data set as 2peaks-manyK (two fODFs peaks and many diffusion kernels). For this, we generated 2 peaks fODFs with crossing angles from 30 to 90 degrees (linearly distributed) and with concentration parameters κ from 8 to 24 (log-linearly distributed), for a total of 65 unique fODF shapes. We generated diffusion kernels with MD from $0.6e-3$ to $1.2e-3$ mm²/s (linearly distributed) and ratios R from 1.5 to 8 (sqrt-linearly distributed), for a total of 30 unique diffusion kernels. We then generated all combinations of fODFs and diffusion kernels, for a total of 1950 different voxel geometries, excluding rotations.

The signal from the fODF and kernel convolution was generated for $N = 60$ b-vectors with a b-value of 1500 s/mm². The data was then overlaid with Gaussian noise, generating data with SNR = [10:10:100] (100 replicates each). To remove potential orientation bias, we added a random 3D rotation to each fODF before generating the synthetic dMRI signal.

Fig. S4.1.2 Different possible ODF geometries for a 75 deg fiber crossing, resulting from different combinations of fODF concentration (i.e., fanning) and diffusion kernels (i.e., microstructural environment). It is easy to see from the figure that different combinations of fODF concentration and kernel shape can result in very similar ODF shapes.

Fig S4.1.2 shows some examples of the range of fODF geometries and diffusion kernel shapes in the 2peaks-manyK simulation dataset. The first column shows diffusion tensors with different ratios R , ranging from 1.5 to 8.0. The first row shows 2 peaks 75 degrees intersecting fODF of different concentration parameters κ , ranging from 24 to 8. The remaining glyphs are the noiseless CSA ODFs corresponding to each pair of fODF and diffusion kernel. These ODFs are used as a proxy to highlight the range of possible signals. The central diffusion kernel represents the median ratio used to generate the 2peaks-oneK data set. We note that it is possible to obtain almost identical ODFs by combining a diffusion kernel with a high R (thin) and a fODF with a low κ (thick), or vice versa. In the case of single-shell diffusion, this fact demonstrates the impossibility of disentangling the concentration parameter of the fODF from the ratio of the kernel. This observation is the main justification for the LSD approach.

Fig. S4.1.3a summarizes the analysis results of the 2peaks-manyK simulation. We reconstructed the fODFs for the 2peaks-manyK dataset ($b=1500$) using LSD and CSD (MRtrix software). The LSD reconstruction used sh order max 8. The CSD kernel estimation was performed using the "dwi2response tournier" method on a specially created companion dataset consisting of the same properties as 2peaks-manyK but with only 1 peak each. We extracted the fODFs peaks and computed the NuFO and angular error. The data were grouped by SNR and crossing angle

for each method. For each bin, we computed the ratio of successes to attempts (STAR metric). Success was defined as a reconstruction with the exact number of peaks as the ground truth and a total angular error of less than 5 degrees. We present the STAR metrics as percentages. We also show the differences between the two methods as percentage points (i.e., a value of 10% in the difference plot means that LSD reconstructed 10% more of the total voxels than CSD for that SNR and crossing angle, not 10% more).

Overall, LSD consistently reconstructs about 10% more total cases than CSD and strongly outperforms CSD in the lower crossing angle regime (less than 60 degrees). This is easily explained by the nature of both methods; while CSD deconvolves the data once with a computed kernel, LSD deconvolves the data with 20 different kernels to obtain 20 candidate fODFs. While the underlying algorithm is different, each of these 20 deconvolutions is still somewhat equivalent to CSD, but LSD then performs a model selection over the candidate fODFs. Therefore, it is not surprising that LSD consistently outperforms CSD as long as the underlying model selection is reasonable.

Simulation Dataset 2peaks-oneK: We generated another 2peaks dataset, which covers the same range of fODF geometries as 2peaks-manyK but employs only a single diffusion kernel. The fODF parameters are sampled more finely within the same range to obtain the same total number of configurations. We refer to this data set as 2peaks-oneK (two fODFs peaks and one diffusion kernel). This microstructural configuration is more consistent with the underlying assumptions of CSD – a single diffusion kernel is used to generate the dMRI signal by convolution.

We generated 2-peak fODFs with crossing angles ranging from 30 to 90 degrees (linearly distributed) and concentration parameter κ from 8 to 24 (log-linearly distributed), for a total of 195 unique fODF shapes. We employed a diffusion kernel with MD of $0.9e-3 \text{ mm}^2/\text{s}$ and ratio R of 4.107, which correspond to the median MD and median R from the manyK dataset. We generated all combinations of fODFs and diffusion kernels for a total of 1950 different voxel geometries, excluding rotations. The signal from the fODF and kernel convolution was generated for $N = 60$ b-vectors and a b-value of $1500 \text{ s}/\text{mm}^2$. The data was then corrupted with Gaussian noise, resulting in $\text{SNR} = [10:10:100]$ (1000 replicates each). We added a random rotation to each fODF before generating the signal to remove potential orientation bias.

Fig. S4.1.3b summarizes the analysis results of the 2peaks-oneK simulation. We reconstructed the fODF for the 2peaks-oneK dataset ($b=1500$) using LSD and CSD (MRtrix software). The LSD reconstruction used sh order max 8. The CSD kernel estimation was performed using the iterative "dwi2response tournier" method on a specially created companion dataset consisting of the same microstructural properties as 2peaks-oneK but with only one peak (i.e., fiber direction). We extracted the fODFs peaks and computed the NuFO and angular error. The data is grouped by SNR and crossing angle for each method. For each bin, we computed the success-to-attempts ratio

(STAR metric). Success was defined as a reconstruction with the exact number of peaks as the ground truth and a total angular error of less than 5 degrees. We present the STAR metrics as percentages. We also show the differences between the two methods as percentage points (i.e., a value of 10% in the difference plot means that LSD reconstructed 10% more of the total voxels than CSD for that SNR and crossing angle, not 10% more).

The experiment with the 2peaks-manyK dataset violated the main hypothesis of CSD, the existence of a unique diffusion kernel. Using the 2peaks-oneK dataset, we show that LSD still outperforms CSD when this hypothesis is perfectly satisfied. The 2peaks-oneK dataset is overall easier to reconstruct, as it does not contain challenging voxel geometries such as low diffusivities and/or low ratios. This results in a higher overall STAR metric for both LSD and CSD. However, the difference map shows the same pattern of LSD slightly but consistently outperforming CSD in the high SNR and crossing angle range, with a very large difference below 60 degrees.

LSD and CSD Simulation Results - Proportion of Correct Reconstructions of Fiber Direction

Fig. S4.1.3: (a) Simulation Results for 2peaks-manyK synthetic tissue. **(Left)** The ratio of successful LSD reconstructions against *in-silico* ground truth for various crossing angles and SNR values. **(Center)** The ratio of successful CSD reconstructions against *in-silico* ground truth for various crossing angles and SNR values. **(Right)** The percentage point difference between successful LSD and CSD reconstructions for various crossing angles and SNR values. Positive values indicate better LSD reconstruction. The simulations show that LSD substantially outperforms CSD reconstruction over a wide range of crossing angles and SNR values. **(b)** Simulation Results for 2peaks-oneK synthetic tissue – considering the main assumption of CSD: microstructural homogeneity. **(Left)** The ratio of successful LSD reconstructions against *in-silico* ground truth for various crossing angles and SNR values. **(Center)** The ratio of successful CSD reconstructions against *in-silico* ground truth for various crossing angles and SNR values. **(Right)** The percentage point difference between successful LSD and CSD reconstructions for various crossing angles and SNR values.

Positive values indicate better LSD reconstruction. The simulations show that LSD substantially outperforms CSD reconstruction over a wide range of crossing angles and SNR values.

In summary, from the two simulation experiments, we can conclude that LSD reconstructions provide substantially higher accuracy of diffusion reconstruction than the established CSD model over a wide parameter range of fiber configuration and data quality. Due to the large variability of *post-mortem* diffusion parameters, we have chosen to perform the simulations using approximately known *in-vivo* human diffusion parameters, without limiting the generality of the results. Our simulation results indicate that LSD can also provide better fiber reconstruction in the human *in-vivo* context.

4.2 Histological Assessment of LSD in Human Sample

We performed a validation of the reconstructed LSD fiber directions by *post-mortem* measurements of a human brain slab. These data allowed us to (i) conclude the validity of the LSD reconstructions based on the known human anatomy and (ii) validate the fiber direction reconstructions using optical histology.

Sample Description

A 2cm coronal human brain slab (89y, 24h PMI) was obtained through the University of Leipzig's body donation program. Brains were provided by the Body donation program, Institute of Anatomy, Medical Faculty, Leipzig University, the Brain Banking Centre Leipzig of the German Brain-Net, operated by the Paul Flechsig Institute of Brain Research and Neuropathology, Medical Faculty, Leipzig University. The entire procedure of case recruitment, acquisition of the patient's personal data, protocols, and the informed consent forms, performing the autopsy, and handling the autopsy material have been approved by the responsible authorities (Approval by the Sächsisches Bestattungsgesetz von 1994, 3. Abschnitt, §18, Ziffer 8; GZ 01GI9999-01GI0299; Approval # WF-74/16, Approval # 282-02 and Approval # 205/17-ek).

The neuropathologic examination did not reveal any evidence of neurologic disease. Following the standard Brain Bank procedures, the sample was immersion-fixed in either 3% PFA and 1% glutaraldehyde in phosphate-buffered saline (PBS) (pH 7.4) for at least 6 weeks. Following it was embedded in 1.5% agarose in PBS for cutting a 50 μm slice at the vibratome (Hyrax V50, Carl Zeiss Jena, Jena, Germany). For immunohistochemical processing, the slice was washed with PBS-Tween 20 trice for 7min. Next, it was immersed in 2% H₂O₂ diluted in 60% MeOH for 60 min and washed for 10min in PBS-Tween 20. Incubation in blocking solution (2% BSA, 0.3% milk powder, 0.5% DNS, 0.1% Na₃in 0.02% PBS-Tween 20) for 60 min followed.

Diffusion MRI Data Acquisition and Reconstruction

The sample was rehydrated and embedded in perfluoropolyether for MRI acquisition. Slab dMRI data were acquired on a preclinical Bruker Biospec 94/20 MRI system at 9.4T (Paravision 6.0.1), using a $G_{\text{max}} = 660 \text{ mT/m}$

gradient system and an 86 mm transmit-receive quadrature radiofrequency coil (Bruker BioSpin, Ettlingen, Germany). Data with an isotropic resolution of 400 μm were acquired using a pulsed gradient diffusion weighted 2D Spin-Echo sequence using the following parameters:

TR = 3074.5 ms, TE = 38.9 ms, matrix size $[r \times p \times s] = 215 \times 175 \times 70$, no partial Fourier, no parallel acceleration, 9 averages. Diffusion-weighting was applied with $b = 10.000 \text{ s/mm}^2$ in 60 directions, on a half-sphere, and partially flipped to equally cover the full sphere. Each repetition contained 4 uniformly interspersed $b=0$ acquisitions without diffusion weighting.

Diffusion data processing entailed (i) removal of data points during the heating phase, (ii) noise characterization and signal debiasing using noise standard deviation and the effective number of coils, (iii) MP-PCA denoising, (iii) Gibbs ringing correction using sub-voxel shift, (iv) linear field-drift correction, (v) correction of eddy currents from diffusion-weighting. Finally, we computed fODFs using both CSD and LSD with $sh = 8$. The CSD computation in MRtrix employed the iterative 'tournier' method (Tournier et al., 2013) to assess a suitable deconvolution kernel. LSD was computed using a series of predefined deconvolution ratios from 1.1 to 10, a relative peak intensity threshold of 0.25, and an angular peak separation threshold of 20 deg. The optimal deconvolution ratio was selected in a $3 \times 3 \times 3$ AIC patch.

Histology Data Acquisition and Reconstruction

For imaging of the histological slice, the Zeiss CLSM (LSM 880 Airyscan, Carl Zeiss Jena, Jena, Germany) equipped with a 20x objective (NA 0.8, WD 0.55mm, RI 1.38, Plan-Apochromat Carl Zeiss Jena, Jena, Germany) was used. Autofluorescence of myelinated fibers (Monici, 2005; Rusch et al., 2022) was excited with an argon laser with 488 nm. The emitted light was collected using 3 band-pass filters: 505–530nm. The image constitutes of 25 x 27 single microscopic images – here referred to as tiles. By deploying the tile scan function of the CLSM, a larger mosaic containing the entire histological slide was acquired. Each tile sizes 120 x 120 μm .

Comparison between LSD and CSD Reconstructions

Direct comparison of CSD and LSD reconstructions based on high-resolution data in known human anatomy (Fig. S4.2.1) suggests similar effects to those observed in the chimpanzee brain sample. Compared to CSD, LSD fODFs appear sharper and more ordered, confirming the known anatomy of the corona radiata. In contrast, CSD fODFs appear blunt and directionally fragmented. This is particularly evident in the inferior part of the corona radiata (Fig. S4.2.1a,b, Magnification 3,4), where the overly sharp CSD kernel estimation resulted in split fODFs, indicating incorrect fiber orientations.

a LSD Reconstruction of Post Mortem Human Corona Radiata**b CSD Reconstruction of Post Mortem Human Corona Radiata**
Fig. S4.2.1 Comparing LSD and CSD fODF reconstruction in a *post-mortem* human corona radiata **(a)** LSD reconstruction alongside four zoomed regions within the corona radiata. **(b)** CSD reconstruction alongside four zoomed regions within the corona radiata. In contrast to CSD, LSD performs adequate reconstructions of the known human anatomy.

Histological Validation of LSD Reconstruction

To enable a comparison between LSD 3D fODFs (Fig. S4.2.2a) and 2D histology (Fig. S4.2.2b), the dMRI fODFs were projected on a 2D space. To do this, the LSD sh coefficients (sh max order 8) of the 3D ODFs were projected onto a fine 3D sphere of 46210 vertices. The deformed sphere was treated as a point cloud and projected onto a 2D plane (coronal surface). The coronal 2D plane was then discretized into 1 deg radial bins (Fig. S4.2.2c). We computed 2D ODFs by finding the point with the maximum norm in the plane for each radial bin. Each 2D ODF was min-max normalized and colored according to its mean weighted direction using the HSV color map on the semicircle.

We computed histology-based fiber orientation maps using microscopic autocorrelation (Wedeen et al., 2012) (Fig. 4.2.2c). For each tile in the histology image, a 2001x2001 crop was selected around the center. This was done to avoid the edges of the tile stitching. For each tile, we then computed the self-correlation using an FFT convolution, normalized by overlap area (100-pixel correlation radius). This central disk was linearly interpolated on a polar grid with radial bins of 0.1 pixels and angular bins of 1 deg. We computed histology ODFs respecting solid angle constraint by multiplying the autocorrelation values by their radius and summing along each radial

bin. Each resulting histology ODFs was then min-max normalized and color-coded according to its mean weighted direction using the HSV color map on the semicircle.

Fig S4.2.2 Comparison of LSD fODF reconstruction and myelin autofluorescence. **(a)** LSD fODFs were reconstructed for a 2 cm *post-mortem* brain slice (left hemisphere) at 400 μm isotropic resolution. **(b)** Myelin autofluorescence data were acquired for a sample of the *post-mortem* slice. 2D histology fODFs were calculated for the histology slice using pixel autocorrelation. **(c)** For comparison with histology, the LSD fODFs **(left)** were converted to 2D by projection onto a coronal plane **(middle)**. The 2D histology fODFs **(right)** for the entire histology sample show similar fiber orientations as detected by LSD. The 2D fODFs are color-coded according to the mean in-plane angle α .

Fig S4.2.2c shows the results of the human corona radiata comparison between LSD reconstruction and myelin histology. The comparison allows us to recognize distinctive features of the fiber structure in the 2D dMRI fODFs as well as in the myelin fiber histology fODFs. For example, the prominent cst (green) and the striatal bridges running transversely to the cst (cyan) are equally visible in both maps. Note that 100% homology between the

two images is not expected because (i) both modalities are sensitive to different tissue properties (i.e., water diffusion vs. myelin), and (ii) the histology data are distorted from the tissue sectioning process.

3. Authors measured 58 uniformly distributed diffusion direction. This is not particularly high angular resolution data compared with available human dataset (e.g., the HCP Young Adult Dataset). I assume that this is because acquisition time is very long because authors used readout segmentation; however, please make explicit statements on why authors made this choice, rather than other choices (such as increasing diffusion direction or performing multi-shell measurements). In addition, considering a limited angular resolution, how much authors are confident about an ability of LSD-fODF to resolve crossing fibers?

Response: The reviewer is correct in pointing out that the number of acquired diffusion directions may be related to the angular resolution of reconstructed fiber orientations. To show that our fiber reconstructions were not affected by a limited number of directions, we used extensive numerical simulations against and *in-silico* ground truth predefined crossing fibers with multiple crossing angles.

To investigate the effect of the number of diffusion directions on our fiber reconstruction accuracy, we generated synthetic dMRI signals between 28 and 200 optimized diffusion directions and compared their reconstruction accuracy with the here employed diffusion reconstructions. To remove averaging effects from the analysis, we balanced the SNR between each synthetic experiment. Our ground truth simulations clearly showed that our acquisition scheme (single shell, 55 diffusion directions) did not compromise the accuracy of the reconstructed fiber orientations. We appreciate the reviewer's valuable suggestion and added our synthetic experiments to the supplements of the manuscript. Based on this, we have now made the choice of the number of directions more explicit, to guide future studies. Please note that we previously described 58 measurements, including 3 b0 images and 55 diffusion directions. We have corrected this typo.

To further investigate the effect of the number of acquired diffusion shells, we compared the LSD fiber reconstruction with a multi-shell reconstruction as implemented in MRtrix. Similar to the simulation comparison between LSD and CSD, we created a multi-shell sampling (2peaks-manyK data set). The synthetic data consisted of 4 b values [700, 1000, 1500, 3000] s/mm² with 60 directions for each shell. The directions were optimized so that each shell was approximately uniform on the sphere and the total of 240 directions projected on the same sphere were also approximately uniform (Caruyer et al., 2013, MRM). Since this multi-shell dataset consisted of 4 times more data than the single-shell method, the SNR per direction was reduced so that the overall dataset was SNR matched ($SNR_{multi} = (1/\sqrt{4}) * SNR_{single}$). We reconstructed the fODFs from the multi-shell data using the MRtrix multi-shell-multi-tissue (MSMT) method. However, we restricted the diffusion kernel generation to only one tissue because this dataset is only simulated WM, and therefore we call the method MSST (multi-

shell-single-tissue). After computation, we compared the MSST reconstructions to the single-shell LSD reconstructions (2peaks-manyK data set).

MSST shows poor reconstruction accuracy on this dataset, being strongly outperformed by LSD in almost all cases (see Fig R1.3). The minimum angular resolution of MSST is high with this diffusion sampling scheme on this dataset. We can clearly see the weakness of the multi-shell approach on this dataset - the multi-shell deconvolution does not include a b-value decay model over multiple b-values. Instead, it simply concatenates all response functions of each shell and solves the global least squares deconvolution problem. In theory, the multi-shell approach could improve fODF reconstruction because the lower b-values have low angular contrast but high relative SNR, while the higher b-values have high angular contrast and low relative SNR. However, in practice, if the SNRs are properly adjusted, it seems that the low angular contrast of the lower b-values overpowers the adjustment process and results in poor fODFs. We conclude from our simulations that the chosen single-shell approach using LSD significantly improves the fiber reconstruction compared to the current state of the art methods, be it single-shell or multi-shell reconstruction. We believe that the additional analyses have strengthened our argumentation and made a stronger case for the chosen approach. We thank the reviewer for this suggestion. As the comparison with multi-shell dMRI is not the subject of the current paper, this comparison is only intended for the review document.

Fig. R1.3: Simulation Results for 2peaks-manyK synthetic tissue. **(Left)** Ratio of successful LSD reconstructions against *in-silico* ground truth for various crossing angles and SNR values. **(Center)** Ratio of successful MSST reconstructions against *in-silico* ground truth for various crossing angles and SNR values. **(Right)** Percentage point difference between successful LSD and MSST reconstructions for various crossing angles and SNR values. The simulations show that LSD substantially outperforms the SNR matched MSST reconstruction over a wide range of crossing angles and SNR values.

Manuscript Changes:

Supplementary Information:

4. Assessment and Validation of Local Spherical Deconvolution

In the following, we describe a more detailed investigation of the accuracy and performance of the LSD based on (i) ground truth *in-silico* simulation experiments as well as (ii) human *post-mortem* dMRI measurements and histology.

To this end, we first investigated the influence of the measured number of diffusion directions on the accuracy of the LSD. We then examined the overall reconstruction accuracy of LSD and compared it to CSD using Monte Carlo simulations for a wide range of fiber configurations and SNR values. [...]

4.1 Numerical Simulation Experiments

Impact of Acquired Diffusion Directions on LSD Reconstruction

To evaluate the impact of the number of diffusion directions on the LSD reconstruction quality, we created two typical crossing fiber geometries, generated the diffusion signal for multiple b-vector sampling, and compared the resulting angular errors. We generated fODFs with both 60- and 90-degree crossing angles (concentration parameter $\kappa = 24$) and a diffusion kernel with $MD = 1e-3 \text{ mm}^2/\text{s}$ and $R = 3$. We then used a b-value of $1000 \text{ s}/\text{mm}^2$ and generate sets of b-vectors for $N = [28:2:200]$ points. The signal from the fODF and kernel convolution was generated for each set of b-vector/b-value. The data was then corrupted with Gaussian noise (1000 noise replicates) for each N, using an appropriate value of σ to match the SNR between sets of b-vectors. The σ was chosen so that \sqrt{N}/σ was constant and $SNR=100$ for $N=55$. We reconstructed fODFs using LSD (sh order max 6) from this noisy data for both crossing geometries for all numbers of diffusion directions N. Finally, we plotted the mean angular error and standard deviation (over the 1000 noise replicates of each N).

Fig. S4.1.1 Angular error of LSD reconstructions for two different fiber configurations as a function of the acquired number of diffusion directions. The solid line indicates the mean error. The dotted line indicates the standard deviation of error.

The data indicate that the reconstruction is insensitive to the number of b-vector directions for the simulated tissue and the diffusion parameters for the 90-degree crossings (Fig. S4.1.1). For the 60-degree crossings, we observed a slightly higher average error and larger standard deviation at the very low b-vector counts, which quickly plateaus around $N=50$ (Fig. S4.1.1). We note that the chosen fODF concentration parameter and diffusion kernel ratio were chosen to be slightly on the sharper side of the potential signal, which is expected to be harder to sample properly at a low number of directions. Given this choice of simulated geometry and the observed plateau, we contend that the $N=55$ directions used in this study is a reasonable choice and we don't expect it to result in any loss of reconstruction accuracy.

4. I do not fully understand a comparison with the Bryant dataset in Supplementary Information. Specifically, I could not follow from where differences in tract trajectory are coming from. While I am convinced that a present study provides higher quality data, Bryant study may use different ROI definition and tractography algorithm. I suspect that a difference in tract trajectory between studies can be mostly explained by differences in ROI definition. How much authors used similar or different placement of ROIs? If authors used similar ROIs as Bryant paper did, how much differences in tract trajectory remain? It is difficult to establish what is a right ROI selection due to a limited

number of previous studies on chimpanzee brains. Therefore, I think that it is difficult to prove that author's ROI selections is superior to those by Bryant et al.

Response: The reviewer correctly points out that differences in the selection of tracking ROIs between the two resources can lead to differences in tractography that are not entirely due to data quality. Indeed, the overlap in tract reconstructions between the two resources changes when Bryant's selected ROIs are applied to the space of the resource presented here. We attribute this largely to the fact that the Bryant ROIs were created with the intention of generating best possible tracts on lower quality data and may not be optimal for the resource presented here. To limit this potential confound between ROI selection and data quality comparison, we have removed the comparison to the Bryant data from the supplements.

5. The National Chimpanzee Brain Resource by Chet Sherwood, William D. Hopkins and Todd M. Preuss (<https://www.chimpanzeebrain.org>) should be mentioned as an existing database of chimpanzee dMRI dataset. The advantage of the present dataset compared with this existing resource should be discussed. Authors used a term "Chimpanzee Resource" in Supplementary material, but this can be potentially puzzling since a term is too similar to the National Chimpanzee Brain Resource.

Response: We agree with the reviewer that the National Chimpanzee Brain Resource is an important dataset and should be acknowledged and discussed as such. We have added text to the Introduction to specifically reference this dataset. Also, as suggested by the reviewer, we have changed the naming conventions in the Supplementary Material to refer to the dataset presented here as the "Eichner resource". In addition, we discuss the advantage of the here presented data.

Manuscript Changes:

Introduction:

[...] As for comparisons with great apes, research is limited to older data, collected before the moratorium (Balezeau et al., 2020; Bryant et al., 2020). Noteworthy in this context is The National Chimpanzee Brain Resource (<https://www.chimpanzeebrain.org>), which provides *in-vivo* neuroimaging data from many chimpanzees. [...]

Specific comments:

6. [Introduction, third paragraph] "Primate brain evolution is characterized not only by a rapid increase in brain size and complexity but also but a marked increase in the proportion of white matter". Authors may consider to cite a review paper from Thiebaut de Schotten et al. *Cortex* (2019; 118, 188-202) which provides an overview of comparative dMRI and also refers chimpanzee studies.

Response: We thank the reviewer for pointing out this informative reference. We cited this review in the suggested sentence of the introduction.

7. [Methods, “Diffusion MRI Tractography”] Authors stated that they used the LSD-fODF and MRtrix for tractography but is this really implemented in MRtrix, or did author implement this processed as a custom code?

Response: The resulting fODFs from LSD use the same spherical harmonics basis functions as the fODFs reconstruction in MRtrix can be readily used for tractography with MRtrix (*tckgen*) using the appropriate threshold for the peak amplitude in the normalized fODFs from LSD. We apologize for not making this clear enough and added this information to the manuscript.

Manuscript Changes:

Methods

Diffusion MRI Tractography

[...] Full brain deterministic streamline fiber tractography was computed based on the LSD fODFs using the MRtrix software *tckgen* (v3.0.3, one seed in each voxel, 4th-order Runge-Kutta integration, step size = 0.125 mm, angular threshold = 60°, relative threshold = 0.25). [...]

8. SLF I reconstruction provided in a supplementary information show two explicit branches; is this an expected pattern or due to a limited ability of resolving crossing fibers?

Response: The reviewer is correct in pointing out that the SLF I appears to show two explicit branches. The thin and heavily crossed fronto-parietal SLF I tract was not reconstructed as one fully connected pathway but appears to be cut into a dorsal and a ventral part. The anatomy of this pathway is not fully described in humans as it cannot be reconstructed easily by blunt dissection. It is assumed to be one continuous flat pathway and the missing part might be completed by a local adaptation of the tracking parameters which was not implemented in this work for consistency and clarity. For better illustration, we provide an image of the apparent two branches of SLF I superimposed on a coronal slide. The ventral branch is located on top of the corpus callosum and is clearly separated from the more medially located cingulum. It may in fact represent a separate branch that has not yet been described in the literature. The current resource will provide the basis for an in-depth analysis of this structure by the community. Without further validation, the apparent segregation should not be interpreted directly.

Fig. R1.8: Tractography reconstruction of SLF I alongside the tract density overlaid on a coronal slice of the colored FA image.

Reviewer 2

Remarks to the Author:

Eichner et al present a high-resolution post-mortem MRI data set of one Chimpanzee monkey brain i.e., great apes that evolution-wise is believed very close to the human brain organization. The authors have collected a high-resolution whole-brain diffusion MRI data set in isotropic 0.5 mm voxel which is a bit higher than the isotropic 0.6 mm image resolution presented by others such as Roumazeilles et al (2020, Plos Biology) and the 1.8 mm by Bryant et al (2020, Plos Biology) though collected in vivo in a larger cohort. The authors did optimize their acquisition parameters (the b-value) for ex vivo dMRI to secure the best contrast to separate crossing fibers. The data set contains a single b-value acquired in 58 directions. The diffusion tensor model (single fiber) and the conventional constrained spherical deconvolution (CSD) multi-fiber model were fitted to the dMRI data set. Also, the authors present the multi-fiber (Local Spherical Deconvolution) LSD model that appears to model fewer apparently spurious fiber directions per voxel than the CSD. The CSD multi-fiber model is an established model with applications in both in vivo and ex vivo data sets. It is unclear why the CSD model when applied to the presented dMRI data set seems to produce spurious fiber directions. Maybe it is related to a kind of degradation of the tissue quality compared to in vivo due to the use of the immersion fixation procedure after a post-mortem interval of four hours. It is well-described that myelin layers normally tightly wrapped around axons are very sensitive to autolytic effects and become loose (Dyrby et al. 2018, Neuroimage). Such myelin loosening effect could be a source of the spurious fiber direction seen in CSD while the presented LSD multi-fiber model may suppress the autolytic effect. Indeed, the authors did check the tissue quality using different staining using light microscopy, but the technique lacks image resolution to the integrity of myelin layers.

Validation of the LSD model using for example a structure-tensor model applied to histology may have shed light on the question. Nevertheless, the anatomical information of white matter pathways revealed with the color-coded diffusion tensor model and tract density maps indicates a high quality of the dMRI data set. Both the major and finer white matter pathways are detected. The virtually traced pathways using tractography appear dense and show fine tract details likely due to a relatively high image resolution compared to the circumference of the tracts - i.e., a low partial volume effect.

Response 1: We thank the reviewer for the valuable feedback and interest in our results. In general, we agree with the reviewer that the high performance of the LSD in comparison to the CSD comes as a surprise and indicates that a single response function for the entire white matter is not appropriate.

To understand whether the observed effect may be due to potential tissue degradation, we performed a series of additional histological experiments, including electron microscopy. In fact, our additional data highlight that

the tissue is of extremely high quality with very well-preserved myelin layers and the observed differences in reconstruction are not due to autolytic effects or loss of myelin layers.

Furthermore, a comprehensive validation and benchmark of our LSD model against CSD was performed by means of (i) numerical comparison against *in-silico* ground truth, (ii) direct comparison using a high-quality *post-mortem* dMRI dataset of well-known human anatomy, and (iii) histological validation of fiber orientation reconstructions. The validation analyses confirmed that LSD's enhanced fiber reconstruction is not attributable to autolytic effects. Instead, we show that LSD's fiber reconstruction significantly improves over CSD due to its locally optimized kernel. Furthermore, the simulations show that LSD generally provides significantly more accurate fiber reconstructions, even under *in-vivo* conditions.

I appreciate the appealingly high quality of the data set and the sensitivity to fine anatomical details emphasized. It is good for the community to share such data sets. However, the scientific impact of providing a single post-mortem data set with only a single standard b-value shell for tractography application is not clear compared to already available *in vivo* and *ex vivo* data sets.

Response 2: Our data provide the first ever detailed look at the non-human primate great ape brain high resolution of 0.5 mm, low image blurring (compared to previous *in-vivo* work presented e.g., in Bryant et al. 2000 with 1.8mm). Our acquired data demonstrate a significant improvement in resolution, by a factor of 47x compared to existing *in-vivo* data, and by a factor of 2x compared to existing *post-mortem* data. In addition, the here presented data for the first time allow to study the complex crossing fiber structure of the ape brain at high resolution (Crossing fibers not resolved in *post-mortem* work by Roumazeilles et al. 2020, see SI for a detailed comparison). High-quality data, as presented here, are of great scientific value for validating connectivity studies along the primate lineage - a topic of increasing scientific interest in recent years (e.g., Liu et al., 2020, Nat. Neur.; Howard et al., 2023, Nat. Comm.; Roumazeilles et al., 2020, PLOS Biol.). We would like to note that the dMRI data upon which the Bryant tract reconstructions rely are not openly available to the public. This fact presents a significant obstacle to further research based on these findings. Conversely, our data set constitutes a completely open and free repository that is available for direct download and use by the scientific community.

The reviewer further inquired regarding the choice of single shell data acquisition. To this end, we have conducted a comprehensive simulation study comparing the reconstruction accuracy of single-shell LSD with multi-shell deconvolution (MRtrix). The simulations impressively demonstrate that our chosen approach - a single optimized diffusion contrast plus LSD reconstruction - allows for much more accurate fiber reconstruction results than

MRtrix multi-shell deconvolution (evaluated based on the number of reconstructed fibers and angular error). We have summarized the results of this multi-shell simulation study also for Reviewer 1, under R1.3.

Manuscript Changes:

Results:

Ultra-High-Resolution MRI Data

[...] Microscopical and histological assessment revealed a well-preserved brain cyto- and myeloarchitecture, as well as well-preserved myelin layers (Dyrby et al., 2018) (See SI). [...]

Discussion:

[...] Our data provide the first detailed access to the complex and fine-grained fiber structure and connectivity of the chimpanzee brain. After previous releases of high-quality *post-mortem* diffusion resources from marmoset monkeys (Liu et al., 2020) and macaques (Howard et al., 2023), our data finally provides a detailed insight into the brain connectivity in our closest living evolutionary counterparts, the great apes. As such, our data complement current evolutionary brain research, by providing the ability to formulate detailed hypotheses and validate findings from large data sources such as the National Chimpanzee Resource. [...]

Supplementary Information:

1. Tissue Quality

Histology and Electron Microscopy Ultrastructure

To assess tissue integrity in more detail, ultrastructural assessment was performed based on electron microscopy data (Fig. S1.2). A small brain slice from the cerebellum (~20mm x 20mm x 5mm) was refixed in 3% paraformaldehyde and 1% glutaraldehyde in 0.2 M sodium cacodylate buffer at pH 7.4 for 48 hours. The refixed plate was sectioned at 70µm using a sectioning vibratome and small triangles (~3mm x 5mm) were cut out. The small triangles were postfixed in buffered 1% osmium tetroxide for 1 hour at room temperature with constant shaking, rinsed in PBS, dehydrated in a graded acetone series with a contrast step of 70% acetone with 2% uranyl acetate, and embedded in Durcupan Araldite casting resin. For structural orientation, semithin sections were cut at 0.5 µm thickness and stained with toluidine blue (Fig. S1.2a). Ultrathin sections (50 nm) were cut on a Reichert Ultramicrotome II. Sections were imaged with a LEO M 912 Omega TEM (Zeiss, Oberkochen, Germany) at 80 kV. Digital micrographs were acquired with a dual-speed 2K-on-axis CCD camera based on a YAG scintillator and processed with the analytical EM software Image SP (TRS-Tröndle, Moorenweis, Germany) (Fig S1.2b-d).

Electron microscopic evaluation of the brain tissue ultrastructure confirmed the relatively high tissue quality observed in classical histology (Fig. S1.1b; Fig. S1.2b-d). Only partially frayed myelin sheaths, shrunken axons, and some tissue voids were observed (Fig. S1.2b-d). This degree of ultrastructural tissue damage is acceptable given the initial fixation of the brain in 4% paraformaldehyde after a four-hour *post-mortem* interval. Based on the assessment of the tissue ultrastructure and the overall integrity of the myelin sheaths, we conclude that the tissue quality is sufficient for high-quality *post-mortem* reconstruction of fiber tracts.

a Regions of Interest Extracted for Electron Microscopy

b WM ROI

c WM ROI Magnified

d GL ROI

Ultrastructural assessment of cerebellum tissue quality using transmission electron microscopy (Zeiss EM 912 Omega) at 80KV

Fig. S1.2: (a) Photomicrographs of a 70µm vibratome section (upper left) showing the region of interest (ROI) extracted for electron microscopical embedding and examination (upper right) and the corresponding toluidine blue stained semithin section (0.5µm) (lower middle). The ROIs used for electron microscopy are indicated. (b) Electron micrograph of white matter ROI from Figures 3 and 6 showing mainly longitudinal cut myelinated fibers. Some partially frayed myelin sheaths and shrunken axons are detectable. Scale bar 5µm. (c) Electron micrograph of white matter ROI from Figures 3 and 6 showing a higher magnification of the mainly longitudinal cut myelinated fibers. Some partially frayed myelin sheaths and few shrunken axons are detectable. Scale bar 500nm (d) Electron micrograph of the granular layer (GL) ROI from Figures 4 and 6 showing cells and nuclei. Some partially frayed myelin sheaths and few tissue voids are detectable. Scale bar 5µm

Major Comments

1. To reduce the post-mortem interval and the risk of introducing the degrading autolytic effects it is unclear why the authors did not perform perfusion fixation on the chimpanzee.

Response: Thank you for your question and there are several reasons a "classic" immersion fixation was used. The first one addresses the technical feasibility since the timepoint and location of extraction could not be planned, as we relied on a specimen sourced from a naturally deceased chimpanzee within a wider network of collaborating zoos and wildlife sites across Europe and Africa. The only feasible option under these conditions is rapid brain extraction and controlled immersion fixation. In addition, perfusion fixation *ex situ* introduces a large number of gas bubbles, resulting in multifocal hypointensity that interferes with MRI (e.g., McKenzie et al., 2022, <https://doi.org/10.17879/freeneuropathology-2022-4368>). Arguably, establishing our method with an immersion fixed brain provides a higher information value to the scientific community since most hominid brain banks curate immersion fixed specimen allowing for a broad application.

2. It is known that chemical fixation reduces the diffusivity in the tissue and the b-value need to be increased accordingly as the authors also did. If the post-mortem tissue quality is high the FA value should be like in vivo values (Sun et al., 2005, MRM). The authors do show a FA map (Fig S3,B) and show a color bar hard to interpret. For the reader to appreciate the data quality the authors should quantify if the FA values in CC are like in vivo values in the litterateur. I would expect the FA values to be like or higher than in vivo (also to in vivo humans) due to the high image resolution hence a lower partial volume effect in CC.

Response: We thank the reviewer for this valuable suggestion. We performed this additional analysis, as requested. To avoid confounding interspecies effects, we compared the cc FA from our sample with *in-vivo* data from the National Chimpanzee Resource. As suggested by the reviewer, the CC FA of our sample was substantially higher, compared to *in-vivo* chimpanzee values, highlighting the high tissue quality of the sample. We have added a section to the SI describing the analysis and results.

Manuscript Changes:

Methods:

Sample Preparation

[...] The tissue quality of the sample was carefully assessed through histology and immunohistochemistry, electron microscopy, and an assessment of diffusion fractional anisotropy (See SI). [...]

Supplementary Information:

1. Tissue Quality:

Diffusion Anisotropy

To further assess the tissue quality of the sample, the fractional anisotropy (FA) of the corpus callosum (cc) was compared against chimpanzee *in-vivo* values. For *post-mortem* samples of high tissue quality, FA values within the cc should be comparable or even higher than *in-vivo* (due to reduced partial volume effects at high resolution) (Sun et al., 2005).

To evaluate this, the FA of the sample was extracted from a region within the cc (Fig. S1.3). The mean FA in the *in-vivo* chimpanzee cc was obtained from the literature. (Phillips and Hopkins, 2012) The here presented *post-mortem* sample had a mean FA of 0.62 ± 0.08 inside the cc. The mean *in-vivo* FA of the chimpanzee cc from literature was 0.49 ± 0.06 . The high DTI values for fractional anisotropy within a homogeneous fiber region are an indication of low tissue degradation and a high tissue quality of the sample.

Fig. S1.3: The FA map of the sample overlaid with the cc region used for the *in-vivo* comparison (displayed in red).

3. The authors should better explain the need to present a new multi-fiber LSD model compared with the standard CSD. It is not clear why a new yet not validated fiber model is to be introduced when the focus is to present a high-quality dMRI data set expected to work with standard fiber models. The fiber model comparison results presented in figure 2 are not convincing. That said, I think the LSD model approach is appealing.

Response: We thank the reviewer for his specific interest in the LSD fiber reconstruction model. We do not believe in an automatism that ensures that new and high-quality data sets can be automatically reconstructed using standard methods. A similar effect was seen with the introduction of 7T human data, for which the existing 3T segmentation algorithms were not applicable. This is because the conventional tools and models were developed based on different data. As a result, high quality data is not necessarily reconstructable with the standard models. In our specific case, this has two main reasons: (i) First, we see anatomical details in the new resolution that are not visible in conventional dMRI data (e.g., the laminae in the cc). We can assume that the microstructural environment in fine structures is different from their surroundings. Models like CSD no longer fit the data because

the basic assumption of CSD - one microstructural environment for the whole brain - is violated. This effect may be less obvious in conventional 3T data, as the lower resolution data are averaged over larger anatomical areas. (ii) We also assume that tissue fixation introduces additional local diffusivity. Again, this violates the basic assumption of the CSD, and again highlights the need for a new model. Based on these reflections, we are convinced that the analysis should be kept in mind, when improving data quality. Therefore, we have decided to bundle the data into one package with a new and improved LSD fiber reconstruction. It is only through this bundling that the full value of the work presented and a novel insight into chimpanzee connectivity emerges. We added this to the instruction of the manuscript.

We agree that additional data would be needed to adequately validate the LSD fiber model. To address the reviewer's point, we have performed an extensive validation of LSD as well as a comparison with CSD. This comparison includes (i) extensive numerical simulations against *in-silico* ground truth (ii) comparison of LSD against CSD based on high quality *post-mortem* human dMRI data, with known anatomy, and (iii) validation of LSD fiber directions using histology. Our new validation experiments underscore that LSD significantly outperforms CSD for crossing fiber reconstruction, even under *in-vivo* conditions. In addition, in the high-quality human *post-mortem* comparison, we also found a high level of spurious fiber directions for CSD compared to LSD. Our validation experiments demonstrate the high accuracy of LSD fiber reconstruction and will help build confidence in the model within the research community. As the additional analyses substantially strengthened the paper, we would like to thank the reviewer for this suggestion. The results of this validation are included in the Supplementary Information.

Manuscript Changes (Data Quality):

Introduction:

[...] To take full advantage of the achieved data quality we have developed a novel reconstruction technique capable of reliably resolving complex microstructural fiber architectures, such as crossing fibers, also in *post-mortem* brain tissue. [...]

Manuscript Changes (LSD Validation):

Please see R1.2 for detailed manuscript changes

4. Line 241: Chimpanzees are monkeys.

Response: We apologize for the unclarity in the text. We were originally referring to the differentiation between monkeys (e.g., Macaques, Marmosets, etc.) and great apes (e.g., Chimpanzees, Gorillas, etc.) We modified the corresponding sentence and hope it is clearer now.

Manuscript Changes:**Results:**

[...] Given that the sfo appears to exist in both evolutionary distant monkeys and as well as chimpanzees, this specific fiber tract may have become redundant in its function in humans. [...]

5. Line 255: "Laminae guide cc axons while growing..." How can a laminae structure act as a guide?

Response: Wiggins et al. 2017 described first the visibility of laminae and septa in structural images of the corpus callosum and their existence in the prenatal brain was clearly demonstrated in previous research (Jovanov-Milosevic et al. 2009, <https://doi.org/10.3389/neuro.05.006.2009>) and they are crucial in brain development. The glial matrix acts to guide axonal fibers to their destinations. This guidance is done by a combination of physical and chemical cues. One of the mechanisms using the lamina is the substrate adhesion: Axons can sense and respond to the adhesive properties of different substrates they encounter during growth. Certain molecules present in the extracellular matrix, such as laminins and fibronectins, can act as adhesive cues for axonal growth. Axons can adhere to these molecules and use them as a physical pathway to navigate towards their targets. This also allows a topographic mapping and allows a spatial order between the target regions in the left and right hemisphere resulting in connections between homologous areas of the two hemispheres.

6. Line 260: The authors say that columnar organization in CC is visible with the diffusion tensor model as well as with the high-resolution FLASH MR data. I do not see any such organization in FLASH MRI data when inspecting figure 3.

Response: We agree with the reviewer that the laminar structure of the cc is hard to make out in the figure. We removed the statement from the manuscript.

7. Line 264 - 271: I do not follow the callosal ratio results presented. Where do the authors show the ratios?

Response: The absolute commissural ratios were not included in the original version of the manuscript. However, the reviewer is right to point out that these figures would be interesting information for the field. We have therefore segmented the cross sections of the main chimpanzee commissures (ac, pc) and calculated the respective ratios to the cc. In the revised version of the manuscript, we have now included these data along with values previously reported for other primate species (Liu et al., Nat. Neuro., 2020). As shown in Figure R2.7, both the ac to cc and pc to cc ratios decrease with brain volume across the primate lineage.

Fig. R2.7: Commissural Ratios for Anterior Commissure and Corpus Callosum as well as Posterior Commissure and Corpus Collosum plotted versus the relative brain size for different primate species. The relative brain volume is normalized to the marmoset. Commissural ratios for other species than chimpanzee have been extracted from Liu et al., Nat. Neuro., 2020

Manuscript Changes:

Results:

[...] Another aspect of data quality becomes evident when observing inter-hemispheric connections, other than the cc. The size ratios of inter-hemispheric brain connections between the cc and the other commissures have shifted throughout primate evolution. The here observed cross-sectional ratio at the brain midline between the ac and cc for the chimpanzee is 0.022 (marmoset: 0.096, macaque: 0.068, human: 0.015) (Liu et al., 2020). The observed cross-sectional ratio at the brain midline between the pc and cc for the chimpanzee is 0.009 (marmoset: 0.060, macaque: 0.016, human: 0.007) (Liu et al., 2020). All ratios are corrected for brain size. [...]

8. Line 272: "... importance to the evolution of brain ...". Are the ratios corrected for brain size differences?

Response: Yes, the callosal ratios corrected for brain size. We have mentioned this in the manuscript (see above).

9. Line 285: Please rephrase the sentence.

Response: We rephrased the sentence.

Manuscript Changes:

Results:

[...] In contrast, the vhc has recently been documented in a *post-mortem* study on the marmoset monkey. This study used a comparable voxel size (corrected for the brain size volume between chimpanzee and marmoset) as

our approach. The observed discrepancy may indicate an evolutionary regression of the vhc, in comparison between apes and monkeys (i.e., not detectable at a comparable resolution between species). [...]

10. Figure 5: Why did the authors not perform Klingler dissection on the brain instead of showing Klingler dissection of a human brain?

Response: We have decided not to perform a Klingler dissection on the present brain, as this procedure would prevent us from performing potential follow-up studies on the brain (MRI and histology). Klingler dissection involves freezing the tissue, which would cause irreparable damage to the tissue - even to the parts that were not exposed.

11. The authors say that the signal-to-noise ratio is high, but it seems that the SNR is not reported.

Response: The reviewer is correct in pointing out that we did not report the SNR in the original submission. We agree with the reviewer that SNR is an important measure of data quality. Therefore, we have included a SNR map of the data in the supplementary information.

Manuscript Changes:

Results:

[...] The data were acquired over multiple days using an optimized segmented 3D EPI sequence, yielding high mean signal-to-noise ratio of 83.6 (See SI), low distortions, and low blurring. [...]

Supplementary Information:

3. Data Quality

3.1 Signal-to-Noise Ratio

We computed a voxel-wise signal-to-noise ratio (SNR) map as the ratio of the first $b=0$ volume and the standard deviation σ extracted from the acquired noise map (Fig. S3.1). The mean SNR of the data over the whole brain is SNR=83.6.

Fig. S3.1: Whole-brain signal-to-noise ratio (SNR) map along with the color-coded histogram. Depending on the brain region and tissue type, the SNR ranges up to 300. The mean SNR of the whole brain is 83.6. The SNR of the WM is about 70.

12. How many times was the diffusion MRI data set repeated and how were the data sets averaged to increase the SNR?

Response: There was no averaging employed in the data acquisition using the optimized, segmented 3D acquisition. The main contributors to boosting SNR during the acquisition are now described in the supplementary information.

Manuscript Changes:

Supplementary Information:

2. Success Factors for High-Quality MRI Acquisition

[...] (iii) Specialized Diffusion MRI (dMRI) Sequence:

To optimize the signal-to-noise ratio (SNR) of the acquisition, we employed a dedicated sequence, designed for *post-mortem* dMRI. Specifically, we leveraged 3D Echo Planar Imaging (EPI) encoding, to generate a higher SNR than typically employed 2D EPI. Further, to increase the signal and reduce distortions (e.g., due to air bubbles) we employed a segmented EPI readout. As a result of this adapted acquisition scheme, no averaging was required to further increase SNR. [...]

13. Line 485: What was the averaged DWI data set used for?

Response: The averaged DWI data set was used for both visualizations showing a clear GM/WM contrast and test-retest analysis. This information was included in the Methods.

14. When performing the retest of the dMRI data set the authors used non-linear registration to align the two data sets and argue it is due to gradient non-linearities of the gradient system. However, a preclinical MRI scanner was used where the gradient non-linearities usually are not a problem. Could the non-linear differences observed be due to mechanical instabilities of the postmortem brain placed in the container? How did the authors ensure the brain is mechanically stable during scanning, storing, and rescanning?

Response: The reviewer is correct in pointing out that mechanical instabilities during scanning may cause nonlinear deformations of the acquired image. To prevent this from happening, we were extremely careful in packaging the sample (see Fig. R3.14). Inside the sample container, we placed sponges against the lateral and ventral parts of the brain. The sample container itself was also heavily padded and mechanically decoupled from the scanner with additional sponges. Additionally, our experiment employs a rather long TR, which also prevents the build-up of vibrational artifacts. Due to the heavy brain padding and the long TR, we did not expect to see any vibration artifacts in the acquisition.

The nonlinear registration between test and retest acquisition was performed to compensate for small deformations occurring during sample placement, as the brain would typically tend to float on top of the immersion oil.

We have added an additional sentence describing the sample padding to the methods section.

Fig. R2.14: Packaging of a brain sample. Packaging inside the spherical sample container included padding on the ventral side and on both lateral sides of the sample. The brain was placed in the scanner with the

ventral side pointing up. During scanning, the spherical container with the brain was additionally padded using foam.

Manuscript Changes:

Methods:

Sample Preparation

[...] Both the brain in the sample container as well as the sample itself were heavily padded with sponges to minimize mechanical coupling between the specimen and the MRI system during data acquisition. [...]

15. The authors mention the challenges of temperature drifts of the brain tissue due to hardware heating of the gradient hardware. It is not clear how the authors practically ensured to reduce such potential temperature shifts. Did they use a constant air-conditioned flow around the tissue? Maybe minor temperature drifts do not matter?

Response: The reviewer is correct in pointing out that temperature drifts from hardware heating may affect the data acquisition. This is particularly true for diffusion-weighted data, as tissue diffusivity is a function of sample temperature. To some extent, tissue heating is beneficial to the measurement as diffusivity, and therefore diffusion contrast, increases with temperature. Direct cooling airflow around the tissue would prevent this beneficial effect of temperature on image contrast and would limit the stabilizing padding of the sample inside the scanner. Therefore, to stabilize temperature, we decided to introduce a dedicated 13h heating phase through an initial dummy phase at the beginning of the experiment. The dummy measurements served the purpose to heat the sample up to a steady state temperature. When measured continuously without interruption, the sample remains at this temperature and show a constant level of diffusion contrast. The dummy measurements were used only to heat the tissue and were not included in the analysis as they do not provide a constant level of diffusion contrast.

The normalized diffusion contrast averaged over the entire sample (Fig. R2.15) illustrates the effect of these dummy scans on the sample temperature. During the heating phase, the dummy scans increase the temperature, resulting in a stronger signal attenuation effect of the diffusion weighting. After the dummy scans, the sample stayed at a constant temperature for the remainder of the measurement. A pilot experiment was performed to determine the required duration of the dummy scans. We changed the manuscript to make this clear.

Fig. R2.15: Heating effect of dummy scans on normalized diffusion signal. The plot displays the normalized diffusion signal averaged over the entire volume. As the temperature increases during the dummy scan period, the diffusivity increases and the diffusion-weighted signal decreases. The dummy scans are utilized to bring the sample to a constant steady state temperature prior to the experiment. This temperature is maintained for the remainder of the experiment, as can be seen from the exponential fit of the data. The fluctuations in the data points are residuals of the directional tissue diffusion contrast that remain after averaging over the entire volume.

Manuscript Changes:

Methods:

MRI Data Acquisition

[...] Before the acquisition, ten diffusion-weighted volumes (almost 13h) were acquired as dummy scans, to achieve a constant steady-state temperature in the sample throughout the scanning session. [...]

Decision Letter, first revision:

Dear Dr. Eichner,

Thank you for submitting your revised manuscript "A Resource for Detailed Mapping of the Complex Fiber Structure and White Matter Pathways of the Chimpanzee Brain" (NMETH-RS49559B-Z). It has now been seen by the original referees and their comments are below. The reviewers find that the

paper has improved in revision, and therefore we'll be happy in principle to publish it in Nature Methods, pending minor revisions to satisfy the referees' final requests and to comply with our editorial and formatting guidelines. Importantly, please make sure that it is very clear that the data are derived from a single subject.

TRANSPARENT PEER REVIEW

ORCID

Best regards,
Nina

Nina Vogt, PhD
Senior Editor
Nature Methods

Reviewer #1 (Remarks to the Author):

The authors have performed a substantial amount of work to address two reviewer's comments. I appreciate the author's effort and think that the manuscript has been improved. I would like to list some minor comments below.

(1) P4, authors wrote: "Due to dated technical acquisition strategies".

I feel that a statement of "dated" is unnecessary. Since all past papers used acquisition strategies at that period, a statement of "dated" does not provide any useful information to readers. The authors just need to state that there is a limitation in the spatial resolution of the existing datasets.

(2) P8, Figure 1. In panels a and c, labels depicting "anterior" in sagittal sections seem incorrect. The label "A" appears on both anterior and posterior sides. In addition, I think that it may be better to define these labels (A, P, L, R, S, I) in the figure legend to help readers.

(3) P14: Authors wrote: "In humans, however, the morphology or existence of the pathways remains controversial³²."

I will suggest authors also cite Forkel et al. (2014, Cortex 56,73-84) which discusses the relevant topic.

(4) P14: Authors wrote: "Given that the sfo appears to exist in both evolutionary distant monkeys and as well as chimpanzees, this specific fiber tract may have become redundant in its function in humans."

I think that the tone of this sentence is too strong, given the fact that the existence of human SFO is controversial. Please consider rephrasing this sentence.

(5) P21: Authors wrote: "do not provide spatial 3D images/representations"

I think that it is not impossible to provide 3D images/representations using PLI (see Palm et al., 2010 Frontiers, Doi: 10.3389/neuro.09.009.2010). The PLI indeed requires sectioning and 3D reconstruction can be a great challenge, but please consider rephrasing this sentence given that there is ongoing effort to enable 3D reconstruction of the PLI data.

(6) P31: Authors note that the segmentation of fascicles is based on guidelines for humans and chimpanzees. I suspect that authors also used macaque guidelines (such as Schmahmann and Pandya, 2006 "Fiber Pathways of the Brain"). If this were true, please include a statement and references to macaque guidelines here.

Reviewer #2 (Remarks to the Author):

The authors have made thorough responses to my questions. Especially, the EM evaluation of the WM is appreciated. However, I suggest the authors emphasize that the sample is taken in the cerebellum where the fixative will quickly penetrate compared to the deep WM region. It might show different results in deep WM, but the comparison with FA in vivo and the author's post-mortem dMRI in the CC is a good indicator of high tissue quality.

For the validation of the LSD fiber orientation model in the supplementary material, it is puzzling to me why the authors use a different diffusion MRI acquisition setup than those used in the manuscript.

Line 372/373 Is not clear to me. What is the problem with using the PLI technique?

In summary, the authors have made a high effort to check the quality of the tissue and they present a fiber orientation method (LSD) that is more robust than the conventional CSD. Still, there exist other MRI datasets on chimpanzees' brains although their voxel size is higher, and some structures do appear clearer. The authors show that major pathways can be extracted as seen in other monkey species and in humans which has interest and is expected. The data set contains only a single b-value shell, so its usage may be limited to tractography analysis of the major WM pathways, but that also has value.

Author Rebuttal, first revision:

Reviewer 1

Remarks to the Author

The authors have performed a substantial amount of work to address two reviewer's comments. I appreciate the author's effort and think that the manuscript has been improved. I would like to list some minor comments below.

Response: We thank the reviewer for the positive evaluation of our work. We are extremely happy appreciative of the reviewer's suggestions which have substantially strengthened the manuscript.

Comments

1. P4, authors wrote: "Due to dated technical acquisition strategies". I feel that a statement of "dated" is unnecessary. Since all past papers used acquisition strategies at that period, a statement of "dated" does not provide any useful information to readers. The authors just need to state that there is a limitation in the spatial resolution of the existing datasets.

Response: The reviewer is correct in pointing out that the term dated does not add substantial value. We replaced it according to the reviewer's suggestion.

Manuscript Changes:

Introduction:

[...] These in-vivo neuroimaging data from chimpanzees, acquired many years ago are limited in image resolution and neuroanatomical accuracy. [...]

2. P8, Figure 1. In panels a and c, labels depicting "anterior" in sagittal sections seem incorrect. The label "A" appears on both anterior and posterior sides. In addition, I think that it may be better to define these labels (A, P, L, R, S, I) in the figure legend to help readers.

Response: We thank the reviewer for spotting this mistake. We corrected it in the final version of the manuscript.

3. P14: Authors wrote: "In humans, however, the morphology or existence of the pathways remains controversial³²." I will suggest authors also cite Forkel et al. (2014, Cortex 56,73-84) which discusses the relevant topic.

Response: We thank the reviewer for this suggestion. We now cite the paper at the suggested location.

4. P14: Authors wrote: "Given that the sfo appears to exist in both evolutionary distant monkeys and as well as chimpanzees, this specific fiber tract may have become redundant in its function in humans." I think that the tone of this sentence is too strong, given the fact that the existence of human SFO is controversial. Please consider rephrasing this sentence.

Response: We toned down the sentence in the discussion.

Manuscript Changes:

Discussion:

[...] Considering that the sfo is substantially reduced, or even absent in humans, the regression of this tract may be an evolutionary brain change specific to humans. [...]

5. P21: Authors wrote: "do not provide spatial 3D images/representations". I think that it is not impossible to provide 3D images/representations using PLI (see Palm et al., 2010 Frontiers, Doi: 10.3389/neuro.09.009.2010). The PLI indeed requires sectioning and 3D reconstruction can be a great challenge, but please consider rephrasing this sentence given that there is ongoing effort to enable 3D reconstruction of the PLI data.

Response: The reviewer is correct in pointing out that PLI may also provide 3D fiber representation. Despite recent progress, 3D PLI remains a highly experimental work in progress. However, the reviewer is right that it is indeed possible to reconstruct 3D fibers from PLI, despite difficulties. We have therefore softened the sentence.

Manuscript Changes:

Discussion:

[...] Other methods for inferring fiber connections, such as invasive *in-vivo* tract-tracing or *post-mortem* polarized-light-microscopy (PLI) (Axer et al., 2011) are either not feasible for chimpanzees (i.e., tracing) or face considerable challenges in the reconstruction of spatial 3D images/representations (i.e., PLI). [...]

6. P31: Authors note that the segmentation of fascicles is based on guidelines for humans and chimpanzees. I suspect that authors also used macaque guidelines (such as Schmahmann and Pandya, 2006 "Fiber Pathways of the Brain"). If this were true, please include a statement and references to macaque guidelines here.

Response: The reviewer is correct that we also consulted the pioneering work of Schmahmann and Pandya as a standard reference. We now reference this macaque literature in the manuscript.

Manuscript Changes:

Methods:

[...] The segmentation of the different fascicles was based on guidelines for humans (Catani and Thiebaut de Schotten, 2008), chimpanzees (Bryant et al., 2020), and macaques (Schmahmann et al., 2009). [...]

Reviewer 2

Remarks to the Author

The authors have made thorough responses to my questions. Especially, the EM evaluation of the WM is appreciated. However, I suggest the authors emphasize that the sample is taken in the cerebellum where the fixative will quickly penetrate compared to the deep WM region. It might show different results in deep WM, but the comparison with FA in vivo and the author's post-mortem dMRI in the CC is a good indicator of high tissue quality.

In summary, the authors have made a high effort to check the quality of the tissue and they present a fiber orientation method (LSD) that is more robust than the conventional CSD. Still, there exist other MRI datasets on chimpanzees' brains although their voxel size is higher, and some structures do appear clearer. The authors show that major pathways can be extracted as seen in other monkey species and in humans which has interest and is expected. The data set contains only a single b-value shell, so its usage may be limited to tractography analysis of the major WM pathways, but that also has value.

Response: We thank the reviewer for the overall positive evaluation of our work – especially regarding the additional data acquisition. The reviewer is correct that the cerebellum may be penetrated with fixative more quickly than the rest of the brain. To point this out, we added an additional sentence to the SI.

Manuscript Changes:

Supplementary Information:

Histology and Electron Microscopy Ultrastructure:

[...] We would like to highlight the possibility that the fixative may penetrate the cerebellum before other brain regions, thereby potentially resulting in higher quality tissue evaluations. To provide an indication of the remaining tissue quality, we performed an dMRI analysis of a highly anisotropic white matter brain structure.

Comments

1. For the validation of the LSD fiber orientation model in the supplementary material, it is puzzling to me why the authors use a different diffusion MRI acquisition setup than those used in the manuscript.

Response: The validation of LSD fiber orientation reconstruction was performed with an adapted MRI acquisition setup to allow for the best possible data quality – through usage of a smaller bore system and a tighter RF coil. As an additional consequence, we expect our data to further support the stability of LSD with respect to the precise choice of dMRI acquisition strategy.

2. Line 372/373 Is not clear to me. What is the problem with using the PLI technique?

Response: The reviewer accurately highlighted that the initial sentence was somewhat misleading, a concern also raised by reviewer 1. In response, we have refined the sentence to enhance clarity and specificity (see R1.5).

Axer, M., Amunts, K., Grässel, D., Palm, C., Dammers, J., Axer, H., Pietrzyk, U., Zilles, K., 2011. A novel approach to the human connectome: ultra-high resolution mapping of fiber tracts in the brain. *Neuroimage* 54, 1091–1101.

Bryant, K.L., Li, L., Eichert, N., Mars, R.B., 2020. A comprehensive atlas of white matter tracts in the chimpanzee. *PLoS Biol.* 18, e3000971.

Catani, M., Thiebaut de Schotten, M., 2008. A diffusion tensor imaging tractography atlas for virtual in vivo dissections. *Cortex* 44, 1105–1132.

Schmahmann, J.D., Schmahmann, J., Pandya, D., 2009. *Fiber Pathways of the Brain*. Oxford University Press, USA, New York, NY.

Final Decision Letter:

Dear Dr Eichner,

I am pleased to inform you that your Resource, "Detailed Mapping of the Complex Fiber Structure and White Matter Pathways of the Chimpanzee Brain", has now been accepted for publication in *Nature Methods*. The received and accepted dates will be July 8th, 2022 and March 29th, 2024. This note is intended to let you know what to expect from us over the next month or so, and to let you know where to address any further questions.

Over the next few weeks, your paper will be copyedited to ensure that it conforms to *Nature Methods* style. Once your paper is typeset, you will receive an email with a link to choose the appropriate publishing options for your paper and our Author Services team will be in touch regarding any additional information that may be required. It is extremely important that you let us know now whether you will be difficult to contact over the next month. If this is the case, we ask that you send us the contact information (email, phone and fax) of someone who will be able to check the proofs and deal with any last-minute problems.

Please note that *Nature Methods* is a Transformative Journal (TJ). Authors may publish their research with us through the traditional subscription access route or make their paper immediately open access through payment of an article-processing charge (APC). Authors will not be required to make a final decision about access to their article until it has been accepted. Find out more about Transformative Journals

If you are active on Twitter/X, please e-mail me your and your coauthors' handles so that we may tag you when the paper is published.

Best regards,
Nina

Nina Vogt, PhD
Senior Editor
Nature Methods